# Searching for the optimal drought index and time scale combination to detect drought: a case study from the lower Jinsha River Basin, China

Javier Fluixá-Sanmartín[1], Deng Pan[2], Luzia Fischer[3], Boris Orlowsky[4], Javier García-Hernández[1], Frédéric Jordan[5], Christoph Haemmig[3], Fangwei Zhang[6], Jijun Xu[2]

[1]Centre de Recherche sur l'Environnement Alpin (CREALP), Sion, 1951, Switzerland
[2]Changjiang River Scientific Research Institute, Changjiang Water Resources Commission, Wuhan Hubei, 430010, China
[3]GEOTEST AG, Zollikofen, 3052, Switzerland
[4]climate-babel, Zurich, 8047, Switzerland
[5]Hydrique Ingénieurs, Le Mont-sur-Lausanne, 1052, Switzerland
[6]Bureau of Hydrology, Changjiang Water Resources Commission, Wuhan Hubei, 430017, China

*Correspondence to*: Javier Fluixá-Sanmartín (javier.fluixa@crealp.vs.ch)

**Abstract.** Drought indices based on precipitation are commonly used to identify and characterize droughts. Due to the general complexity of droughts, the comparison of index-identified events with droughts at different levels of the complete system, including soil humidity or river discharges, rely typically on model simulations of the latter, entailing potentially significant uncertainties.

The present study explores the potential of using precipitation based indices to reproduce observed droughts in the lower part of the Jinsha River Basin, proposing an innovative approach for a catchment-wide drought detection and characterization. Two indicators, namely the Overall Drought Extension (ODE) and the Overall Drought Indicator (ODI), have been defined. These indicators aim at identifying and characterizing drought events at basin scale, using results from four meteorological drought indices (Standardized Precipitation Index, SPI; Rainfall Anomaly Index, RAI; Percent of Normal precipitation, PN; Deciles, DEC) calculated at different locations of the basin and for different time scales. Collected historical information on drought events is used to contrast results obtained with the indicators.

This method has been successfully applied to the lower Jinsha River Basin in China, a region prone to frequent and severe droughts. Historical drought events occurred from 1960 to 2014 have been compiled and catalogued from different sources, in a challenging process. The analysis of the indicators shows a good agreement with the recorded historical drought events at basin scale. It has been found that the time scale that best reproduces observed events across all the indices is the 6-month time scale.

## 1 Introduction

Drought is a natural phenomenon that results from persistent lower precipitations than what is considered as normal. It generally affects larger areas than other hazards and more people than any other natural catastrophe (Keyantash and Dracup, 2002; Wilhite, 2000).

In China, droughts represent the most severe natural threat for socioeconomic development and ecosystems (Mei and Yang, 2014). Drought events occur in the Jinsha River Basin (JRB) and surrounding regions with high frequency. They affect a wide range of areas and cause huge losses to the agriculture sector (He et al., 2013). The clustering of severe and sustained droughts in southwest China during the last decade has resulted in tremendous losses, including crop failure, lack of drinking water, ecosystem degradation, health problems, and even deaths (Wang et al., 2015).

To reduce and anticipate such drought impacts, a comprehensive characterization of the phenomenon is essential to which effective and accurate analysis of hydrometeorological data is a key input. Drought indices are useful for tracking droughts and providing a quantitative assessment of the severity, location, timing and duration of such events (World Meteorological Organization and Global Water Partnership, 2016), but also for real-time monitoring (Niemeyer, 2008), risk analysis (Hayes et al., 2004) and drought early warning (Kogan, 2000).

Some organizations and agencies already rely on the use of indices in their decision-making processes, thus enhancing proactive drought management policies (Wilhite, 2000). An example is the U.S. Drought Monitor (USDM, 2017), an index-based drought map that policymakers use in discussions of drought and in allocating drought relief. Other platforms such as the European Drought Observatory (Joint Research Centre, 2017), the China's Department of Climate Change, National Development and Reform Commission (Department of Climate Change, National Development and Reform Commission, 20   2017) or the experimental African Drought Monitor (Land Surface Hydrology Group - Princeton University, 2017) also use this approach for the assessment, diagnosing and forecasting of droughts.

The choice of index should be based on the type of drought (meteorological, agricultural, hydrological or socio-economical), the climate regime and the regions affected, as well as the available data. It was found that measured meteorological data was limited in the study region and that precipitation was the single most reliable type of exploitable information. The 25   present study thus focuses on the use of meteorological indices based only on precipitation data. The main advantages are their ease of use, the limited data requirements and the capacity for early detection of drought events, while extensive literature and calculation tools are widely accessible (World Meteorological Organization and Global Water Partnership, 2016).

The use of integrated indices such as PDSI (Palmer, 1965) or SPEI (Vicente-Serrano et al., 2010), relying also on potential 30   evapotranspiration (PET) data, could improve the scope and quality of this study. However, no reliable PET data was accessible for the study region. Although approximations may be applied to estimate this variable, for example, by only considering temperature data, some studies (Jeevananda Reddy, 1995; Shaw and Riha, 2011; Staage et al., 2014) showed a high sensitivity of the PET to the chosen equation. A deeper analysis that helps selecting and applying such methods should

be performed prior to the use of these indices. Therefore, it has been decided to base this study on the Standardized Precipitation Index (SPI, McKee et al., 1993a, 1995), the Rainfall Anomaly Index (RAI, Van Rooy, 1965), the Percent of Normal precipitation (PN, Barua et al., 2011) and the Deciles (DEC, Gibbs and Maher, 1967).

To fill the lack of specific drought-related information, most studies assess the performance of drought indices against results

from hydrological soil water models (Halwatura et al., 2016; Hao and AghaKouchak, 2013; Trambauer et al., 2014; Vasiliades et al., 2011; Wanders et al., 2010). However, the performance of these types of studies depends on the accuracy of the models. Their limitations and uncertainties represent an important drawback and should be addressed (Mishra and Singh, 2011). An alternative that often requires more time-consuming work is the compilation of historical records of drought events from different sources. Consequently, their duration, the water scarcity levels and the drought impacts on population

and agriculture can be estimated and then integrated into the analysis. This enables one to identify other types of droughts such as socio-economical droughts that are hard to assess with hydrological models.

Regarding their spatial resolution, the available drought indices may be based on local measurements (Zhou et al., 2012) and index calculations are usually applied to stations or cells of gridded precipitation datasets; overall spatial patterns at catchment or sub-catchment scales are thus hardly captured. As stated above, droughts affect large areas whose limits are

often vaguely demarcated. Besides, water resources are part of a more complex interrelated network which links the source to the point of consumption, where isolated rainfall deficiencies do not necessarily imply a shortage of water availability or even a drought event. Some work (Bhalme and Mooley, 1980; Fleig et al., 2011; Mitchell et al., 1979) suggests the use of drought area indices for the study of droughts that considers areal coverage. The use of overall indicators capable of capturing in a single value the effect of the rainfall deficiency at a regional level is thus convenient and will be applied in this

study based on the above-mentioned work.

The objective of this study is to capitalize on the collection of drought events that the authors have registered in the lower part of the JRB since 1960 to evaluate and calibrate two indicators capable of identifying drought occurrence and characterizing their intensity at catchment scale. These indicators are based on commonly used meteorological drought indices for particular time scales.

**2 Investigation area and data**

The JRB is a sensitive zone in terms of water resources, food security, ecosystem management and human well-being where glacier and climatic variability greatly influence the water regimes and availability. Originating from the southern glacier at Jianggendiru peak, the highest point of the Geladaindong Snowy Mountain in the middle of the Tanggula Mountains, the JRB constitutes the upper part of the Yangtze River Basin. It is located between 24°28′N–35°46′N longitude and 90°23′E–

104°37′E latitude in southwestern China, with a catchment area of 473,200km$^2$ (Fig. 1). The total length of the river is 3'500 km from Yibin city, with a total fall of 5,100 m. This part of the Yangtze River accounts for 55.5% of its length and 95% of its total fall.

The lower part of JRB is a hot-dry valley region characterized by a southwest monsoon climate. The hydrologic regime is characterized by a pronounced seasonal cycle with an annual average precipitation of 600–800 mm/year. Dry season (November to April) precipitation accounts for 10% to 22% of the annual precipitation. Evaporation is 10 to 20 times of precipitation during the dry season, which could be the major reason for the frequent occurrence of winter drought or winter–

spring droughts in lower JRB (Mei and Yang, 2014; Yang et al., 2013). Droughts occurring in the lower JRB and surrounding areas affect a wide range of areas, causing huge losses in agriculture (Wu et al., 2011): more than 4 million people and 3 million livestock face drinking water shortage, and more than 1 million hm$^2$ of cultivated area are susceptible to be affected by severe droughts and water shortages, with expected direct economic losses of hundreds of millions USD (Wu, 1999).

Figure 1 shows the division of the JRB in three parts (Upper, Middle and Lower), and the locations of the meteorological stations used. This study focuses on the analysis of drought events in the lower JRB. The precipitation data needed in this study have been obtained from the China Meteorological Data Service Center (CMA), and downloaded from its data sharing service system (CMDC, 2017). A preliminary quality check and correction of datasets (including data gap-filling) is performed by CMA before uploading them to the system. The monthly precipitation data of 29 meteorological stations

within or around JRB, recorded from 1960 to 2014, have been collected and processed. More than 50 years of continuous data are thus available, except for the Batang and Yuanmou stations where only 46 years are available. The spatial distribution of the stations is supposed adequate for the purposes of the study: the stations are distributed relatively evenly both in the zonal and meridional directions, with no zones having a significantly denser presence of stations that could overestimate their importance.

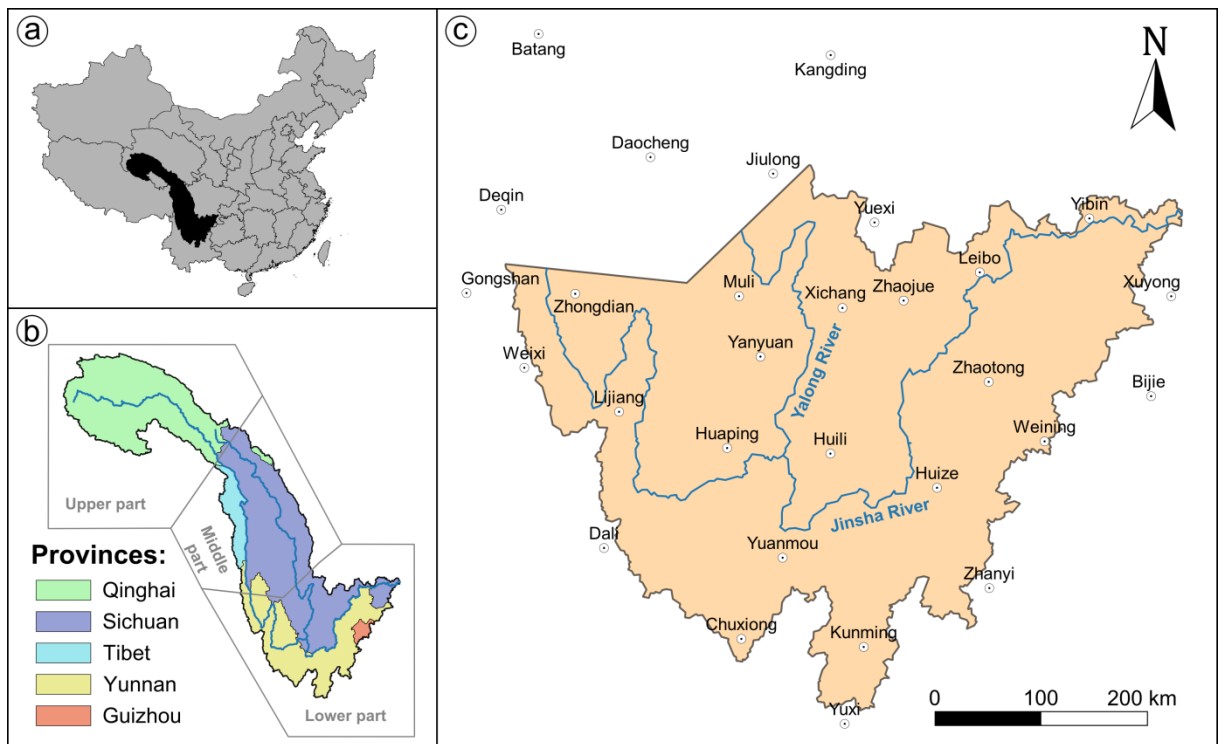

**Figure 1.** a) Location of the JRB in China; b) subdivision of the JRB for this study; c) overview of the lower JRB with the location of the 29 meteorological stations.

## 3 Catalogue of historical droughts

In order to obtain a good basis for the evaluation of drought indices performance, historical drought events have been collected since 1960. The information required for the identification and characterization of major droughts in the lower JRB has been compiled from different sources, including scientific literature, inventories (e.g., international disaster database, Chinese inventories), governmental reports and yearbooks, newspaper and internet articles.

Detailed information is available for the major drought events over the past 20 years. Before 1980, much less information

about droughts in the lower JRB is available. Moreover, detailed information prior to 1960 could not be found.

Compiling and harmonizing the information from these different data sources was a challenge. The drought event descriptions in the scientific literature often give an overview about the entire event in a descriptive way without detailed information about the affected area and damage. They often provide information about the meteorological conditions and the duration of the event. Available information from government reports and databases, in contrast, generally contain

information in high detail for a specific county (e.g., affected areas in km$^2$), but neither give information about the entire affected area if the event affected several counties, nor about prevailing meteorological conditions.

Information at very different levels of detail and various contents were collected. As a first step, all available information was registered in a database. For this purpose, a web-based event registration platform and database (GEOTEST AG, 2017) has been developed to provide a standardized analytical framework with a quantitative description of the drought characteristics. Particularly, drought events since 2000 are all mentioned in a different source, what significantly enhances

the reliability of their existence and related information. Most of the drought events before 2000 are documented in detail in He (2010), and some of these major dry periods are also mentioned in scientific literature such as He et al. (2016) and Wang et al. (2015). Even if the amount of data and level of detail is lower for these older events, their occurrence and temporal positioning can be assumed as reliable.

In a second step, the level of detail was harmonized for the most relevant information and summarized in Table 1. This

catalogue of the most relevant drought characteristics focuses on the affected area, the start date of the drought events, their duration, their spatial and temporal distribution, their severity and the impacts on the population and agriculture, including damage and financial losses. The time of occurrence and duration is given in seasonal units, the affected area is described on a county scale and the indicators and impacts are summarized in a descriptive way, as better accuracy was not feasible for all events.

In total, 13 major drought episodes have been registered from 1960 until 2014. However, this data set is probably not complete, as non-documented events likely have occurred. A clustering of severe and sustained droughts in the JRB has been observed from 2009 to 2014. Another period with high drought activity and severity can be detected between 1980 and 1990. Although the droughts identified from 2009 until 2014 were extremely serious, this was not the worst period in the long-term because the drought episodes that occurred around 1940 were of similar intensity and duration (Wang and Chen, 2012). The

registered drought events are often strongly correlated with low precipitation, but the analyses also reveal that the registered droughts often occur during periods with temperatures above average.

**Table 1.** Catalogue of historical drought collected for the lower JRB (DJF=December-January-February, MAM=March-April-May, JJA=June-July-August, SON=September-October-November).

| ID | Year | Seasons | Affected area | Reference | Other indicators | Impacts |
|----|------|---------|---------------|-----------|------------------|---------|
| I | 1962–1963 | SON, DJF, MAM, JJA | Yunnan, southern part of Sichuan | He, 2010 | Precipitation deficit of 50% from November 62 to April 63. | Drinking water shortage for 900 000 people. Impacts on 3 700 km$^2$ agricultural land. |
| II | 1978–1979 | DJF, MAM, JJA | Yunnan | Liu, 2012 | | Impacts on 7 000 km$^2$ agricultural land, poor harvest/crop loss. |
| III | 1981–1982 | DJF, MAM, JJA | Sichuan, northern Yunnan | He, 2010 | | Drinking water shortage for 2 million people or 3 million people and 2 million livestock. |
| IV | 1987 | exact duration unknown | Mainly Yunnan | | | Impacts on 6 000 km$^2$ agricultural land, poor harvest/crop loss. |
| V | 1992 | MAM, JJA, SON | Yunnan, southern Sichuan | He, 2010 | Maximum in precipitation deficit: 50–80%. | Drinking water shortage for 2 million people and 1 million livestock. Impacts on 9 300 km$^2$ agricultural land. |

| | | | | | | |
|---|---|---|---|---|---|---|
| VI | 1998–1999 | DJF, MAM | Mainly Yunnan | He, 2010 | Temperatures in Yunnan province 2–3°C higher than long-term average. Dayao County: 150 days without rain. | 8 000 km² damaged agricultural area. |
| VII | 2000–2001 | DJF, MAM | Sichuan, Yunnan | WCB, 2001 | Temperatures in Yunnan province 2–3°C higher than long-term average. The cities of Dali, Baoshan, Dehong, Chuxiong, Lincang have almost no rainfall during the whole winter. | Drinking water shortage for 3 million people and 2 million livestock. Impacts on 5 800 km² agricultural land. |
| VIII | 2005 | MAM, JJA | Large parts of Yunnan | Yang et al., 2012; Liu Yu et al., 2007 | High temperatures; in April to early June, the temperature is 1°C above the same period of history in most parts of Yunnan province. Precipitation deficit of 20–80% in May-June November. 56 days without precipitation. | Drinking water shortage for 6 million people and 4 million livestock. Impacts on 15 200 km² agricultural land, poor harvest. |
| IX | 2009–2010 | SON, DJF, MAM | Parts of Yunnan, Sichuan and Guizhou | Yang et al., 2012; Wang et al., 2015 | Precipitation deficit. 119 days without precipitation. Average temperature anomaly of plus 1°C. | Drinking water shortage for 21 million people and 11 million livestock. Impacts on 43 500 km² agricultural land, poor harvest. |
| X | 2011 | MAM, JJA, SON | Large areas in southwest China | Yang et al., 2012; Wang et al., 2015 | Temperatures 0.4–1.1°C higher than normal. From June to September 2011, persistent high temperature weather conditions. Precipitation deficit of 20–60%. | Drinking water shortage for 12 million people and 9 million livestock. Impacts on 19 000 km² agricultural land. Cargo shipping has been suspended. |
| XI | 2011–2012 | DJF, MAM | Large areas in southwest China | Wang et al., 2013 | Precipitation deficit. | Drinking water shortage for 2.4 million people and 1.6 million livestock. Impacts on 6 500 km² agricultural land. |
| XII | 2012–2013 | Oct-Apr | Southwest China | Guha-Sapir et al., n.d.; Hu Xueping et al., 2015 | From October to April 0.5°C higher temperatures than normal, in February 2.5°C higher than long-term average. Jan-Feb: precipitation deficit of 45–55%. | More than 3 million people and about 2 million large livestock had drinking water shortage with varying degrees. 323 small rivers and 331 small reservoirs dried up. 23 300 km² agricultural area affected (whereof 15 500 km² forest). |
| XIII | 2014 | DJF, MAM | Central Yunnan and south Sichuan | Duan et al., 2015 | Spring temperatures 2–4°C higher than historic values in SW China. Spring precipitation in central Yunnan and south Sichuan province was 50–90% less than average of the same period. | Drinking water shortage for 1.6 million people in Yunnan province. 106 rivers and 76 reservoirs dried up. Affected area: 6080 km². |

## 4 Meteorological drought indices

Four different commonly used meteorological drought indices have been applied in this study: the Standardized Precipitation Index, the Rainfall Anomaly Index, the Percent of Normal precipitation and the Deciles. Their definition basically rests upon the comparison of precipitation values with normal value (the definition of "normality" may vary from one index to another),

5 resulting in a single number. This allows characterizing drought conditions and thus facilitating its interpretation and use in strategic planning and operational applications (Tigkas et al., 2013).

This comparison must be month or season specific. For instance, for the index calculation of January 2000, the precipitation of this month should be compared to the normal precipitation extracted taking into account only the Januaries from a reference period. The same applies when calculating the index for the time window January-February-March 2000: the

10 precipitation for these 3 months will be compared to the sum of precipitation of all the groups of January-February-March registered in the reference period.

## 4.1 Standardized Precipitation Index (SPI)

The widely used Standardized Precipitation Index (SPI) was formulated by McKee et al. (1993a, 1995) to quantify the precipitation deficit from long-term recording and for multiple time scales.

Long-term record of precipitation values is fitted to a probability distribution which is then transformed into a standard normal distribution, of which mean and variance are 0 and 1, respectively (Edwards and McKee, 1997). The data sets are most commonly adjusted to the Gamma function (McKee et al., 1993a; Sönmez et al., 2005; Tsakiris et al., 2007) although some studies show better adjustments to other functions (Akbari et al., 2015).

A classification of drought conditions based on the SPI values was established by McKee et al. (1993a) to define drought intensities and is presented in Table 2. Positive SPI values indicate greater than normal precipitation, and negative values indicate less than normal precipitation.

**Table 2.** Classification of drought conditions according to the SPI values.

| SPI | Classification |
|---|---|
| $\geq 2.0$ | Extremely wet |
| 1.5 to 1.99 | Very wet |
| 1.0 to 1.49 | Moderately wet |
| –0.99 to 0.99 | Near normal |
| –1.49 to –1.0 | Moderately dry |
| –1.99 to –1.5 | Severely dry |
| $\leq -2.0$ | Extremely dry |

As mentioned earlier, the SPI was designed to quantify precipitation deficit for multiple time scales or moving time windows (World Meteorological Organization, 2012). These time scales reflect the drought impacts on different water resources which are needed by decision-makers:

- 3-month SPI: reflects short- and medium-term moisture conditions and provides a seasonal estimation of precipitation.
- 6-month SPI: indicates seasonal to medium-term trends in precipitation and may be very effective in showing the precipitation anomaly over distinct seasons. Information from a 6-month SPI may also be associated with anomalous streamflow and reservoir levels, depending on the region and time of year.
- 12-month up to 24-month SPI: reflects long-term precipitation patterns and is usually tied to streamflow, reservoir levels, and even groundwater levels at longer time scales.

## 4.2 Rainfall Anomaly Index (RAI)

The Rainfall anomaly Index (RAI) was developed by Van Rooy (1965). The RAI indices are computed by comparing the average precipitation over a given time window with the mean of the ten highest (for positive anomalies) and the ten lowest

(for negative anomalies) precipitation records. Despite its simplicity, this index requires a series of complete data to be calculated.

The RAI values are classified (Van Rooy, 1965) as shown in Table 3. Olukayode Oladipo (1985) found that differences between the RAI and the more complicated indices of Palmer Drought Index (Palmer, 1965) and Bhalme and Mooly Drought Index (Bhalme and Mooley, 1980) were negligible.

Table 3. Classification of the period according to the values of the RAI.

| RAI | Classification |
|---|---|
| ≥ 3.00 | Extremely wet |
| 2.00 to 2.99 | Very wet |
| 1.00 to 1.99 | Moderately wet |
| 0.50 to 0.99 | Slightly wet |
| –0.49 to 0.49 | Near normal |
| –0.99 to –0.50 | Slightly dry |
| –1.99 to –1.00 | Moderately dry |
| –2.99 to –2.00 | Very dry |
| ≤ –3.00 | Extremely dry |

## 4.3 Percent of Normal precipitation (PN)

The percent of normal precipitation (PN) is one of the simplest measurements of precipitation value for a location. It is calculated by dividing precipitation during a given time window by normal precipitation of that same time window over the reference period (typically considered to be a 30-year average). For PN values over 100%, the precipitation is higher than the average precipitation (and vice versa): the higher PN value, the wetter the considered month is.

The main advantage of this index is its simplicity and transparency, which makes it practical to communicate drought levels to the public (Keyantash and Dracup, 2002). Analyses using PN are very effective when used for a single region and/or a specific season.

Even if no threshold ranges have been widely established in the technical literature for the PN, some studies (Barua et al., 2011; Morid et al., 2006) propose a classification similar to the SPI. For this study, the classification proposed by Barua et al. (2011) has been adopted (Table 4).

Table 4. Classification of drought conditions according to the PN values.

| PN | Classification |
|---|---|
| 180% or more of normal rainfall | Extremely wet |
| 161% to 180% of normal rainfall | Very wet |
| 121% to 160% of normal rainfall | Moderately wet |
| 81% to 120% of normal rainfall | Near normal |
| 41% to 80% of normal rainfall | Moderately dry |

| PN | Classification |
|---|---|
| 21% to 40% of normal rainfall | Severely dry |
| 20% or less of normal rainfall | Extremely dry |

## 4.4 Deciles (DEC)

Another drought-monitoring technique consists in dividing the monthly precipitation data into deciles (DEC). This method, developed by Gibbs and Maher (1967), was selected as the meteorological measurement of drought for the Australian Drought Watch System (Lee, 1979; Sivakumar et al., 2010) because it is relatively simple to calculate and requires less data and fewer assumptions than the Palmer Drought Severity Index (Smith et al., 1993). The procedures have also been adopted by the World Meteorological Organization to monitor drought on a worldwide scale (World Meteorological Organization, 1985).

The threshold ranges of deciles used to classify drought conditions are presented in Table 5 (Gibbs and Maher, 1967).

**Table 5.** Classification of drought conditions according to the values of the deciles.

| DEC | Percent | Classification |
|---|---|---|
| Deciles 1–2 | lowest 20% | Much below normal |
| Deciles 3–4 | next lowest 20% | Below normal |
| Deciles 5–6 | middle 20% | Near normal |
| Deciles 7–8 | next highest 20% | Above normal |
| Deciles 9–10 | highest 20% | Much above normal |

## 5 Approach for the identification of drought events at basin scale

An indicator (or indicators) capable to adequately characterize historical droughts must be able to capture the following characteristics:

- ▪ The beginning and the end of the event, which defines its duration.
- ▪ The drought intensity, derived from the index value.
- ▪ The geographical area affected by the drought.

The following guidelines specify the approach proposed in this study to characterize drought events at basin scale based on precipitation data available at each station and how to contrast these results with the catalogued historical events.

First, following the previous definitions (Sect. 4), precipitation data are used to calculate the four above-described meteorological drought indices (SPI, PN, RAI and DEC) for each station and for different time scales (1-, 3-, 6-, 12-, 24- and 48-month) using the 1951-2000 reference period. Then, according to the criteria presented below, these values are used to detect potential drought events at a given station and at a given time.

In order to aggregate results from all stations of the basin, two indicators are proposed in this study: the Overall Drought Extension (ODE) and Overall Drought Indicator (ODI). The results of these indicators will then be contrasted with historical

recorded events to define the best combination of index and time scale used for the definition of the ODE and ODI indicators.

## 5.1 Use of indices to detect droughts at station scale

According to McKee et al. (1993), a drought event occurs at the station level any time the SPI is continuously negative and the SPI reaches a value of −1.0 or less, which corresponds to moderately dry condition (Table 2) or drier. The drought begins when the SPI first falls below zero (mean of the normalized precipitation) and ends with the positive value of SPI following a value of −1.0 or less. The drought magnitude is the positive sum of the SPI for each month during the drought event. The intensity of a drought is defined as the magnitude of this event divided by its duration.

Figure 2 shows an example of the SPI-6, SPI-12 and SPI-24 series calculated at the Chuxiong station. Drought periods are colored in orange and the lower threshold that defines their occurrence in red. The influence of the time scale on the number and duration of detected droughts is clearly apparent. It is worth noting that there are periods in Fig. 2 that identify very short droughts (one or two months long), which is due to the identification criteria based on the index values.

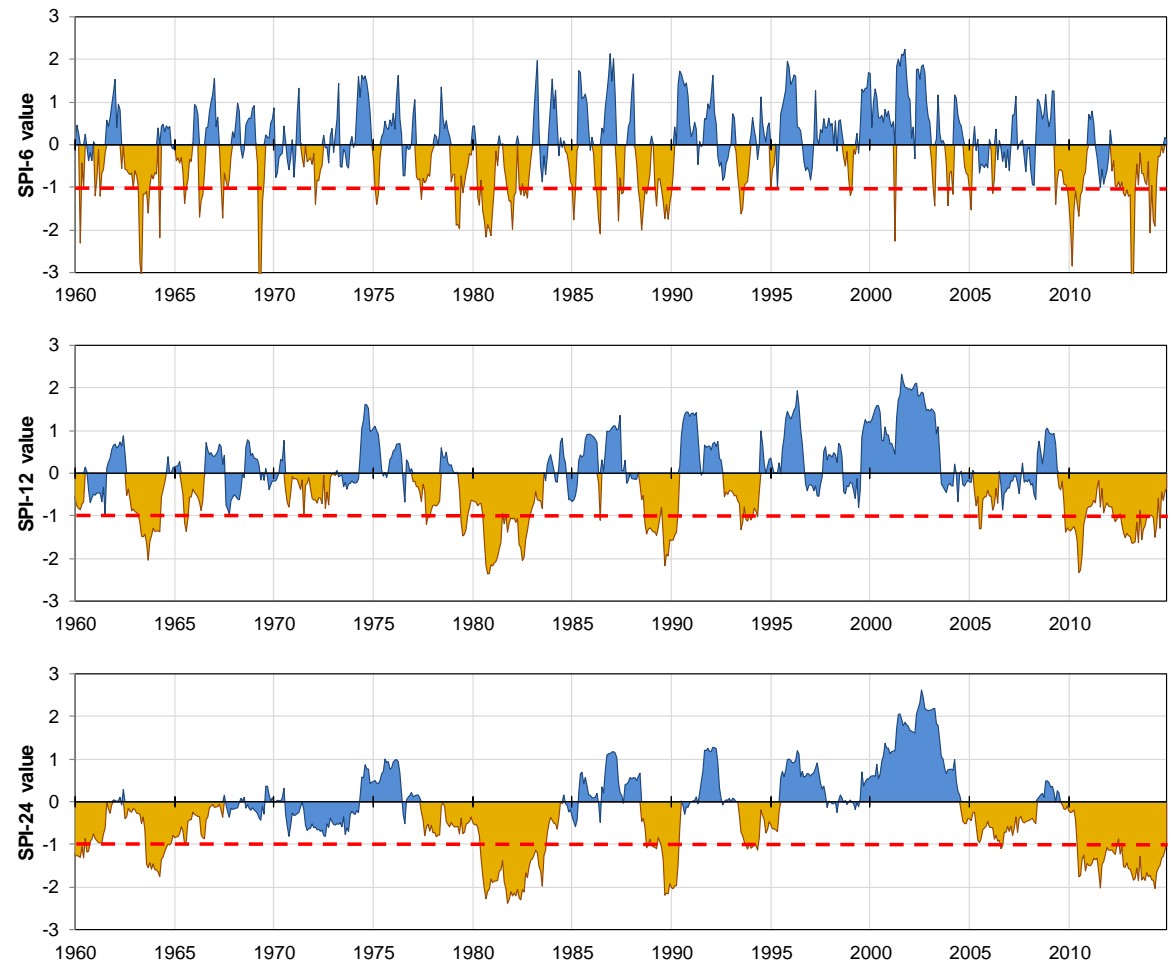

**Figure 2.** Example of the SPI-6, SPI-12 and SPI-24 series at the Chuxiong station, indicating drought periods in orange (lower threshold in red).

5    In the present study, the above-mentioned principles used to detect drought events based on the SPI classification (McKee et al., 1993b) have been standardized to be applicable to the other three indices (PN, RAI and DEC) as follows:

- A drought event occurs any time the index is continuously below its normal value and reaches the moderately dry condition class.
- The drought is considered to begin when the index first falls below its normal value.

10    ▪ The drought ends when the index exceeds its normal value.

Table 6 summarizes the thresholds for each index that specify the drought event's start and end criteria, which correspond respectively to the limit of the moderately dry class and to the index normal value. Although the "normal value" of DEC would be 50% (which corresponds to the median of the precipitation records), in this study the drought end criterion

suggested for this index is 60%, which is the limit between near and above normal conditions (Jain et al., 2015; Tsakiris et al., 2007).

**Table 6.** Values of the thresholds defining the start and the end of the drought events, for each index.

| Index | Start (moderately dry condition) | End (normal value) |
|---|---|---|
| SPI | −1 | 0 |
| RAI | −1 | 0 |
| PN | 80 | 100 |
| DEC | 40% | 60% |

## 5.2 Identification of drought occurrence

When analyzing directly these meteorological indices, results only concern each station's surroundings without capturing the patterns of neighboring areas. However, available historical records refer to regional droughts characterized by larger areas that cover several stations.

In order to consider the basin as a whole in the definition of drought occurrence, duration and intensity, the resulting indices must be consistently extended to the entire area and then combined in overall indicators. For that purpose, a regular grid divides the lower JRB into a 400x300 cells raster (400 rows and 300 columns). The grid resolution was chosen as a trade-off between the density of cells (1 cell/3.2 km$^2$) and the computational requirements (however, this choice should be adapted to the needs of potential other cases). After trimming off the areas sticking out of the basin boundaries, the raster possesses 44 133 cells. Index values have been calculated at each grid cell by applying the Inverse Distance Weighting (IDW) spatial interpolation from the values available at the stations.

For this study, we followed the approaches taken for the definition of Drought Area Indices (Bhalme and Mooley, 1980; Mitchell et al., 1979) and Regional Drought Area Indices (Fleig et al., 2011). It is thus considered that a basin-wide event is ongoing when a substantial part of the basin is under drought conditions. It is therefore necessary to identify the portion of the territory for which the calculated index indicates a drought. An indicator to detect drought occurrence at basin scale has been set up based on the criteria described above to identify an event considering the index values (Fig. 2).

Based on the interpolation of the index, drought events are detected for each time step and at each grid cell of the described raster. This allows us to define an indicator, named here Overall Drought Extension (ODE), expressed as the percentage of the lower JRB area suffering a drought. It is calculated as the number of cells indicating a drought at a precise date (N_drought) divided by the total number of cells of the raster (in this case, N_TOTAL=44 133) as shown in Eq. (1),

$$ODE = \frac{N\_drought}{N\_TOTAL} \cdot 100\%. \tag{1}$$

The ODE ranges from 0% (when no drought is occurring at any point of the basin) to 100% (when the entire basin is suffering an event). It highlights the coverage of a drought, allowing a direct comparison between registered historical

information and calculated results. Moreover, it helps define the temporal component of droughts as it states the beginning and the end of an event. However, it does not take into account its intensity.

## 5.3 Characterization of drought intensities

Regarding the intensity of the droughts, in this study a complementary indicator is applied to integrate the intensities
computed at every grid cell. The Overall Drought Indicator (ODI) is defined as the average index value across the cells under drought conditions at a precise date, as shown in Eq. (2),

$$ODI = \frac{\sum_{i=1}^{N\_drought}(Index_i)}{N\_drought} \, . \tag{2}$$

The ODI expresses the average severity in the drought-affected part of the basin. It gives information about the meteorological stress level of the areas being effectively affected by a drought. Moreover, this indicator may help complete
the collected historical records which include little information on the magnitude of the events. From indications of Table 2, Table 3, Table 4 and Table 5, lower values of this indicator denote drier conditions. Undefined values occur when no cells are under drought conditions.

Only cells under drought conditions have been considered to define this indicator. If the ODI had been calculated as an average value for the entire basin (as adopted for instance in Trambauer et al. (2014)) higher (or lower) indicator values in a
part of the basin may have compensated lower (or higher, respectively) indicator values in the rest of the basin, yielding an overall value close to normal precipitation. Therefore, the ODI must always be used together with the ODE: whenever a drought has been detected with the ODE, its overall intensity may be assessed with the corresponding value of the ODI.

## 5.4 Evaluating indicator-based results with catalogued historical events

In order to support the choice of an index and time scale combination for the definition of the ODE and ODI, an assessment
of the quality of the forecasts performed with the different variants is recommended. The hypothesis is that detected drought events (i.e., the forecasts) correspond to the cases when the ODE value exceeds a given threshold, which indicates a certain area is affected by an event. The temporal coincidence of these forecasts has to be then contrasted with the occurrence of recorded droughts (i.e., the observations). As stated above, Fig. 2 shows that very short index-based events risk being forecasted. In order to avoid an overestimation of droughts, an additional 3-month criterion for beginning and ending
forecasted droughts was established: an event will be effectively detected when the ODE value exceeds the threshold for at least three consecutive months.

For the matching between forecasts and observations, two monthly series of events were created (one for the events detected according to the ODE values; and one for the historical events), where for each month either a "Drought" or a "No drought" condition is assigned. Different scores for contrasting this type of dichotomous forecasts (occurrence vs. no occurrence)
exist: the Peirce skill score, PSS (Hanssen and Kuipers, 1965; Murphy and Daan, 1985; Peirce, 1884); the Heidke skill score,

HSS (Heidke, 1926); the Gilbert's skill score, GSS (Schaefer, 1990); or the odds ratio skill score, ORSS (Stephenson, 2000). As recommended by Candogan Yossef et al. (2012), the PSS is used in this study. For its calculation, the Miss Rate (M) and the False Alarm Rate (F) are defined in Eq. (3) and Eq. (4) respectively:

$$M = \frac{c}{a+c} \, , \tag{3}$$

$$F = \frac{b}{b+d} \, , \tag{4}$$

where a, b, c and d represent the number of cases for each possible forecast outcome, respectively (Table 7):

- hit: when one detected drought corresponds with an observed drought;
- false alarm: when a drought appears during a month where no observed event has occurred;
- miss: when, during a month where a drought has been observed, no event has been detected;
- correct rejection: when, during a month where no drought has been observed, no drought is detected.

The Miss Rate (M) indicates how many of the observed events are not forecasted (related to the Type 1 errors) while the False Alarm Rate (F) is the proportion of non-occurrences that are incorrectly forecasted (Jolliffe and Stephenson, 2003). The PSS is expressed as shown in Eq. (5):

$$PSS = 1 - M - F \, , \tag{5}$$

The PSS ranges from −1 to +1: perfect forecasts receive a score of one, random forecasts receive a score of zero, and negative values indicate less skill than a random prediction. A suitable combination of the index and time scale will then lead to higher PSS values.

**Table 7.** Contingency table of the comparison between forecasts and observations.

| | | Observation | |
|---|---|---|---|
| | | **Drought** | **No drought** |
| **Forecast** | **Drought** | a (hit) | b (false alarm) |
| | **No drought** | c (miss) | d (correct rejection) |

However, high values of the PSS score may be obtained purely by chance, especially when using only a small number of forecasts. Such is the case of the present work, where only 13 independent events have been documented during the 55 years of record keeping. This could lead to overestimating the goodness of a combination of the index and time scale. A statistical test was applied to check if the calculated PSS values were significantly different from zero, at least at a 95% confidence.

Assuming independence of the Miss and False Alarm rates, the standard error in the Peirce skill score is simply the square root of the sum of the squared standard errors in the Miss and False Alarm rates (Stephenson, 2000), as expressed in Eq. (6):

$$SE_{PSS} = \sqrt{(SE_M)^2 + (SE_F)^2} \, , \tag{6}$$

where the standard errors in estimated Miss ($SE_M$) or False Alarm ($SE_F$) rates can be extracted from Thornes and Stephenson (2001). If the $PSS \pm 1.96 \cdot SE_{PSS}$ interval does not include zero, then the null of a random forecast can be rejected at a 95% confidence level.

## 6 Results and discussion

Following the previous approach, the series of the SPI, PN, RAI and DEC indices were calculated for different time scales (1-, 3-, 6-, 12-, 24- and 48-month) within the period of 1960–2014. First computed at the 29 stations, these indices were then extrapolated to the rest of the lower JRB. Figure 3 shows the example of the Standardized Precipitation Index for a 6-month time scale (SPI-6) calculated in October 2012 (corresponding to the drought event XII) and spatially distributed at the lower JRB, where brown colors represent regions under drier conditions.

According to the criteria proposed in Table 6, detected drought events were identified based on the index values. Then, the ODE and ODI indicators were calculated for the lower JRB. An example of the resulting ODE and ODI series is shown in Fig. 4 for the SPI-6 and RAI-6 combinations and in Appendix A for all the time scales and indices analyzed, along with the recorded historical droughts shaded in orange.

The objective is to establish a combination of time scale and index that offers an optimum identification of historical

droughts. As stated before, the main criteria used to contrast the performance of the forecasts is that a drought event is supposed to happen when the ODE value exceeds a threshold that is to be defined. The combination finally retained should maximize the number of hits and minimize the misses between the forecasts and the observed events.

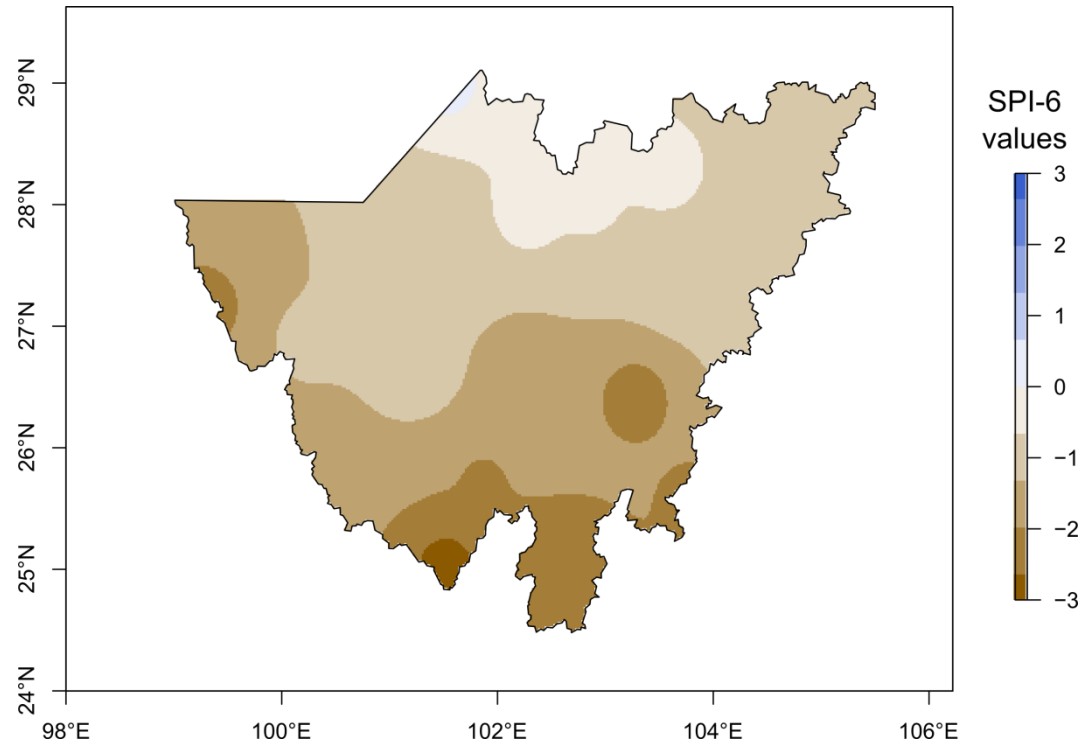

**Figure 3.** Extrapolated SPI-6 values in October 2012 for the entire lower JRB.

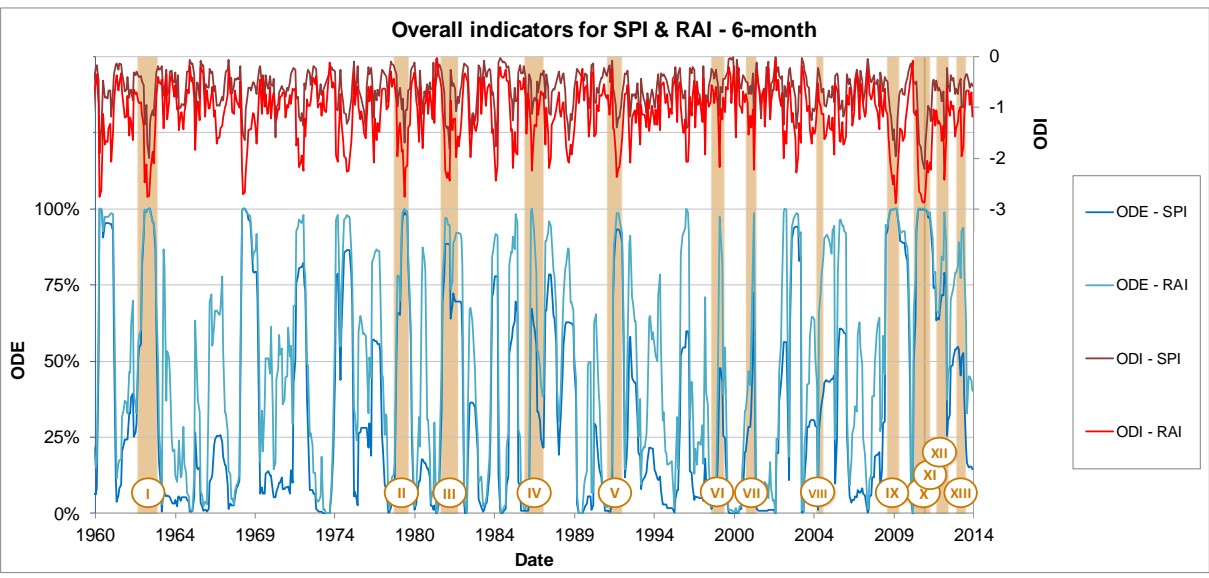

**Figure 4.** ODE and ODI values using the 6-month time scales of SPI and RAI indices, compared with the 13 detected historical droughts (in orange).

The 1-month scale overall indices show rapid fluctuations that correspond to short periods of precipitation deficiency not captured in the catalogue of historical droughts. This is mainly due to punctual large rainfall events, that have an important influence in the indices which may indicate that the drought had ceased when it is not the case (Barua et al., 2011). The use of this time scale is not recommended for drought monitoring since long drought events are hardly identified. The opposite effect occurs when using the 48-month scale. The inertia of the rainfall shortage tendencies may mask shorter droughts and overestimate their durations. Since most of the episodes last one year or less (Table 1), they are hardly detected using the 48-month scale. The droughts which occurred from 2009 to 2014 (droughts IX to XIII) illustrate this phenomenon: even if five different droughts have been catalogued, a unique one is detected using the 48-month scale, according to the ODE time series. Therefore, using the 1- and 48-month scales do not provide any substantial information about the occurrence and duration of the droughts and have been excluded from the performance analysis.

For the rest of the time scales (3-, 6-, 12- and 24-month), the ODE threshold indicating the occurrence of a drought is required for the computing of the PSS that will serve as a support for the selection of the best combination of index and time scale. Traditionally, cross-validation techniques are used to define optimum thresholds, for when within a training subset the threshold maximizing the PSS is identified and validated in a non-overlapping validation subset. The limited number of 13 independent events in our record prevents following this approach. Instead, a sensitivity analysis was performed using the same threshold across all of the combinations and exploring the effect of varying it in a reasonable range (in this case, from 0.3 to 1 by 0.1 steps). The resulting PSS values are shown in Fig. B1 of Annex B along with the 95% confidence interval, which allows indicating whether the score is significantly different from zero.

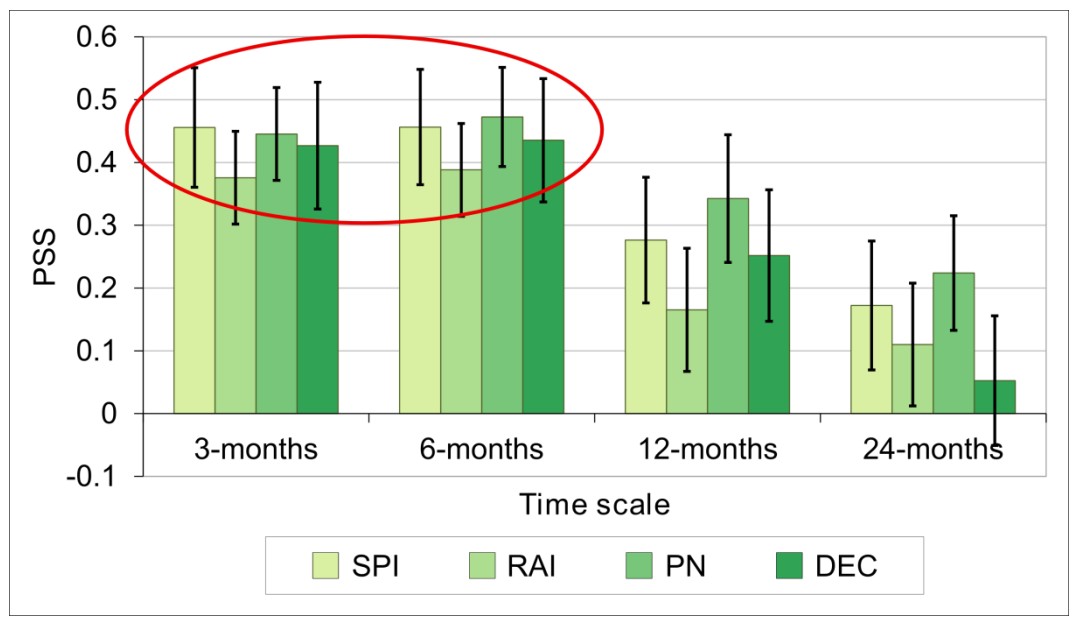

**Figure 5.** Graphic representation of the PSS results for an ODE threshold of 0.4, with the black error bars representing the 95% confidence interval (±1.96 standard errors).

Most of the 95% confidence intervals of the PSS do not include zero, disproving that skill scores could have identified drought events by chance sampling fluctuations. Only for some of the indices at 24-month (e.g., RAI-24 for an ODE threshold=0.7), results cannot assert that skill scores are significantly different from zero and thus these combinations should not be considered.

Attending to the PSS values (Fig. B1), results show a consistent tendency across all ODE thresholds of higher PSS at the 3- and the 6-month time scales. An example of PSS for an ODE threshold of 0.4 is presented in Fig. 5. Moreover, there is no single index that clearly produces better results. Indeed, based on the PSS values and taking into account their confidence intervals, there are no statistically significant differences across the different indices for the 3- and 6-month time scales. This indicates that, for these time scales, all the indices perform similarly well on capturing the events, which is consistent with the fact that they all rely on the same type of data (precipitation). PSS results are independent of the specific threshold and thus they are considered robust. However, it is worth mentioning that in general, higher PSS for the 3- and the 6-month time scales are produced using ODE thresholds between 0.4 and 0.6.

Regarding the 6-month ODE series (Fig. A5 and Fig. A6 of Appendix A), it is important to highlight some relevant aspects:

- All the observed drought events have their corresponding ODE peaks.
- Although event VIII has an estimated duration of 3 months, ODE and ODI results consistently show a longer drought. The exact period of this drought is not well defined as indicated in the catalogue, leaving room for a longer duration of the real episode.
- In general, all the indices are well correlated, identifying most of the recorded droughts.
- Several droughts are consistently detected between event I (1962) and II (1979) even if no drought has been chronicled (false alarms). This may correspond to the above-mentioned scarcity of reliable information on droughts prior to 1980.
- The drought events IX, X, XI, XII and XIII are well captured. As shown in Fig. A5 and Fig. A6, the different events during this period (2009–2014) match with the consecutive increases in the ODE values for all the indices (DEC, PN, RAI, SPI).
- However, the 6-month series of ODE suggest some false positive detections: more drought events than the observed are calculated.

Regarding the 3-month ODE series (Fig. A3 and Fig. A4), results suggest an overestimation of the number of detected events, as sometimes several detected events combine into one (longer) observed event. The 6-month time scale appears as more appropriate.

In summary, according to the ODE series presented in Appendix A, and to the forecast verification carried out with the Peirce skill score (Appendix B), it seems that the best time scale for the identification of droughts is at 6-months. Results show an equally effective performance of the ODE series for all the indices. However, the risk of false positives must be addressed carefully, as the observation record likely misses events, in particular between 1962 and 1979.

Despite the good performance shown by the overall indicator ODE to detect droughts, caution is advised. In particular, the choice of meteorological indices as a basis for the calculation of the ODE and ODI can lead to errors when assessing drought occurrence. Temperature variability, not considered here, can play a significant role in the onset of agricultural drought. Meteorological indices may not be fully capable to capture the impacts on water scarcity and could be complemented with other types of indices, such as agricultural or hydrological. The same approach proposed in this study is recommended using more comprehensive indices in order to better capture the complex drought processes.

The performance assessment of the ODE indicator to detect droughts relies basically on the comparison with the historical events catalogued in this study. The search and compilation of this information from different data sources, often scarce and ambiguous, represents a challenge. Different information sources often provide only partial information for one episode, and for some events the differences in the available information complicate the harmonization of data. As a result, the accuracy of the collected information may impact on the applicability of the developed methodology.

## 7 Conclusions

This study aims at applying overall drought indicators representing the drought status within the entire lower JRB investigation area. This work represents an attempt at building a tool for drought monitoring and risk management purposes at basin scale. It is based on established meteorological indices for the identification of droughts and a method for a catchment-wide drought assessment and characterization, which is compared to historical drought events of the lower JRB.

The information used for the identification and characterization of major historic droughts were compiled from different sources. A total of 13 major droughts between 1960 and 2014 were identified in the lower JRB and catalogued using a web-based registration platform, allowing for a comparison of the different events.

Drought indices typically assess local water deficits while available historical records usually refer to regional droughts. To overcome this problem, two drought area indicators, the Overall Drought Extension (ODE) and the Overall Drought Indicator (ODI), have been used to characterize the occurrence and intensity of an event within a specific investigation area. These indicators are based on four common meteorological indices at different time scales: the Standardized Precipitation Index (SPI), the Rainfall Anomaly Index (RAI), the Percent of Normal precipitation (PN) and the Deciles index (DEC). By relying exclusively on precipitation, the proposed procedure serves as a basis for further studies in other regions where only precipitation data is available.

The performance of the ODE at detecting droughts has been assessed by contrasting the results of this indicator with historical recorded events, offering promising results. It seems that the best results are independent of the index used and produced using the 6-month time scale. Although results suggest the same patterns for all ODE thresholds, it has been noticed that highest PSS values are produces for thresholds between 0.4 and 0.6, which can be defined as a trigger to detect the occurrence of a drought in the lower JRB.

Considering the challenge that the compilation of historical drought information represents and the identified limitations, this is a good method for the monitoring of drought episodes within an entire catchment. The use and contrast of drought indicators at basin scale with historical collected information represent the main innovative aspects of this study. Since meteorological droughts are the first stage in the progression of subsequent agricultural or hydrological droughts, this methodology could be used to activate a management response for a drought event, which starts at a specific threshold value. Additionally, this methodology can be used to complete lacking information on droughts' duration, geographical extension or intensity.

## 8 Appendices

### Appendix A: series of ODE and ODI indicators

The series of the Overall Drought Extension (ODE) and the Overall Drought Indicator (ODI) have been calculated for the 1-, 3-, 6-, 24 and 48-month time scales. Graphic results are presented in Fig. A1 to Fig. A12 below.

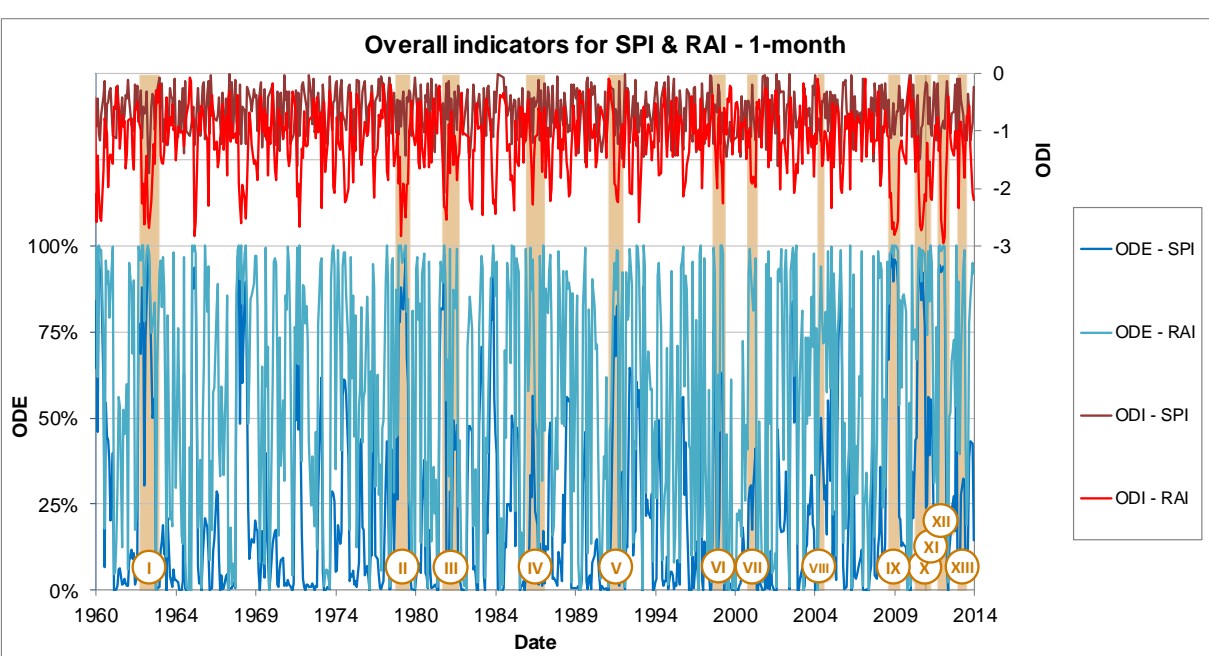

**Figure A1.** ODE and ODI values using the 1-month time scales of SPI and RAI indices, compared with the 13 detected historical droughts (in orange).

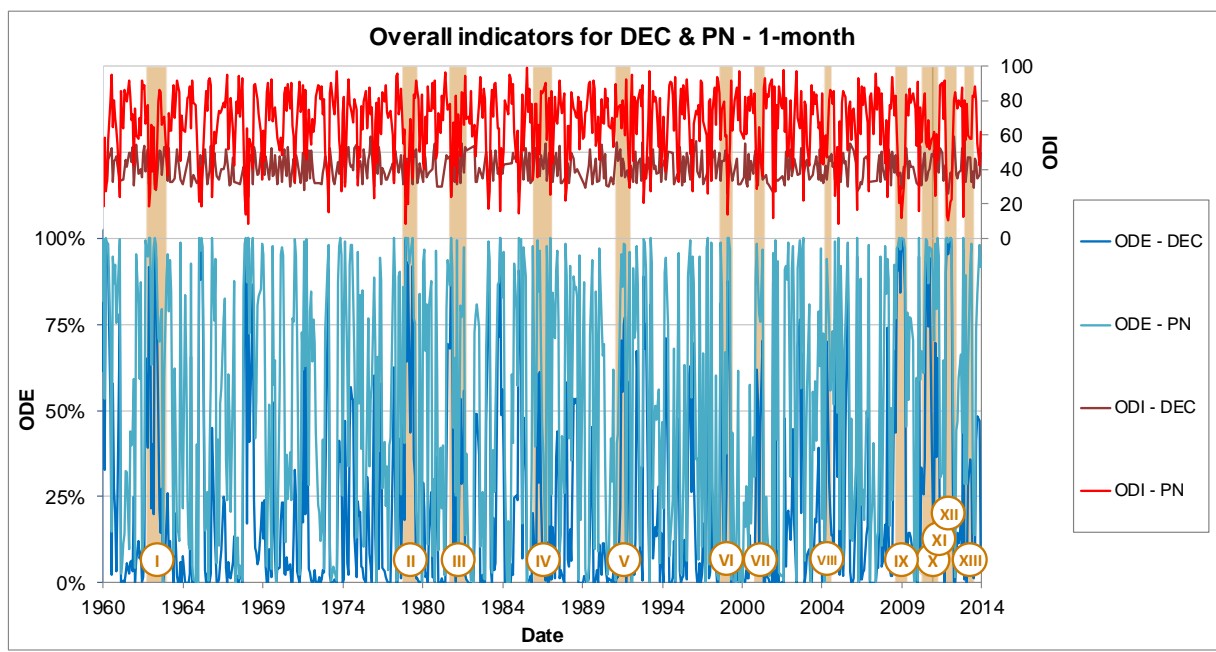

**Figure A2.** ODE and ODI values using the 1-month time scales of DEC and PN indices, compared with the 13 detected historical droughts (in orange).

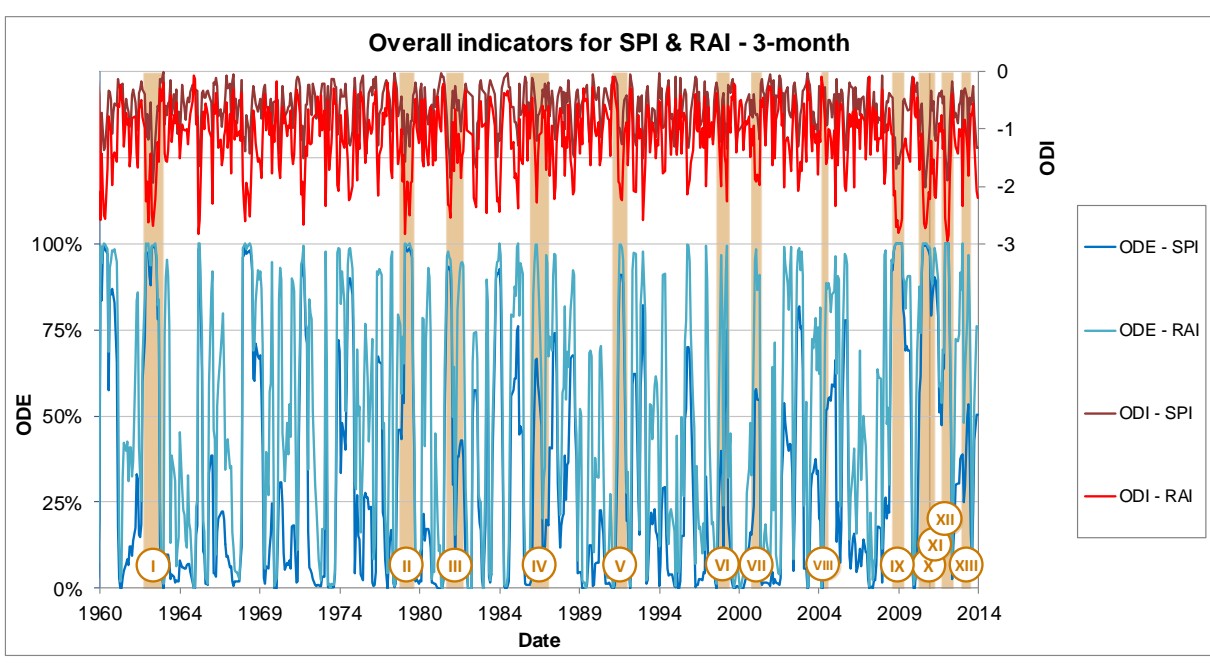

**Figure A3.** ODE and ODI values using the 3-month time scales of SPI and RAI indices, compared with the 13 detected historical droughts (in orange).

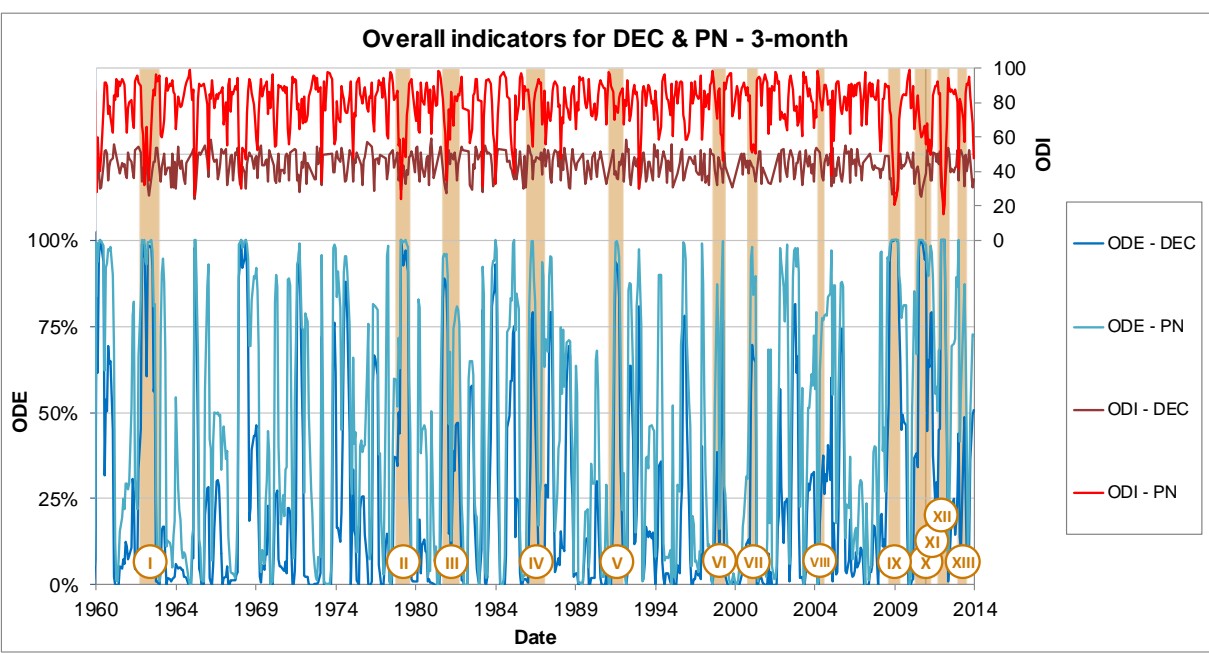

**Figure A4.** ODE and ODI values using the 3-month time scales of DEC and PN indices, compared with the 13 detected historical droughts (in orange).

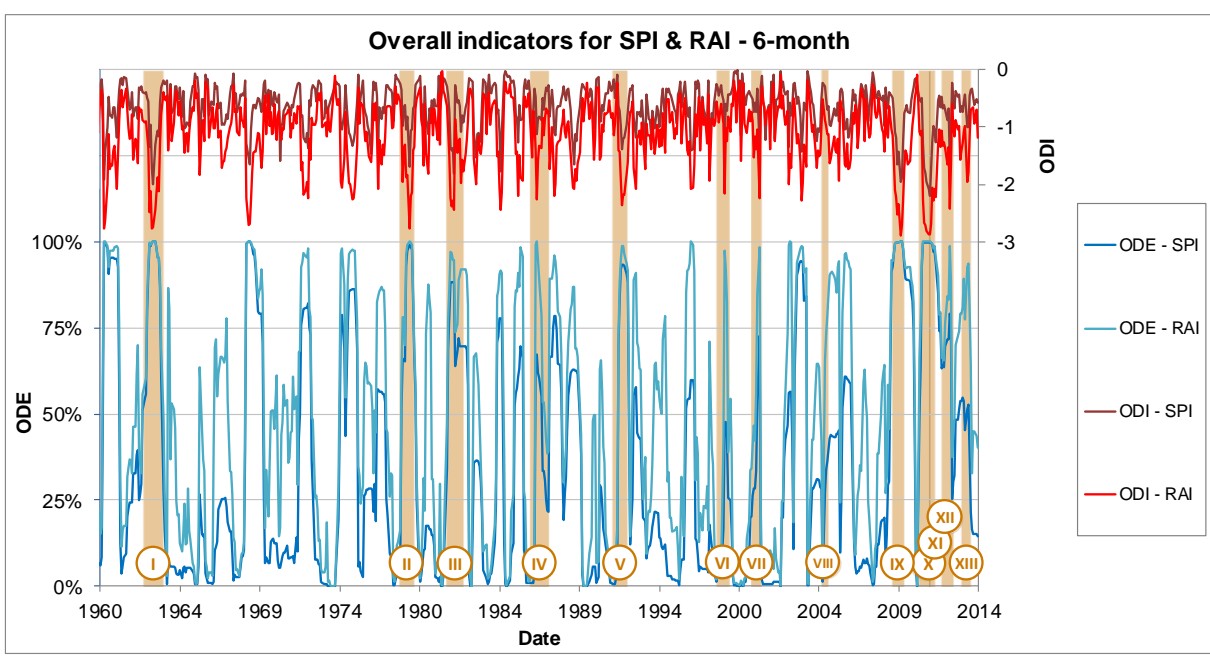

**Figure A5.** ODE and ODI values using the 6-month time scales of SPI and RAI indices, compared with the 13 detected historical droughts (in orange).

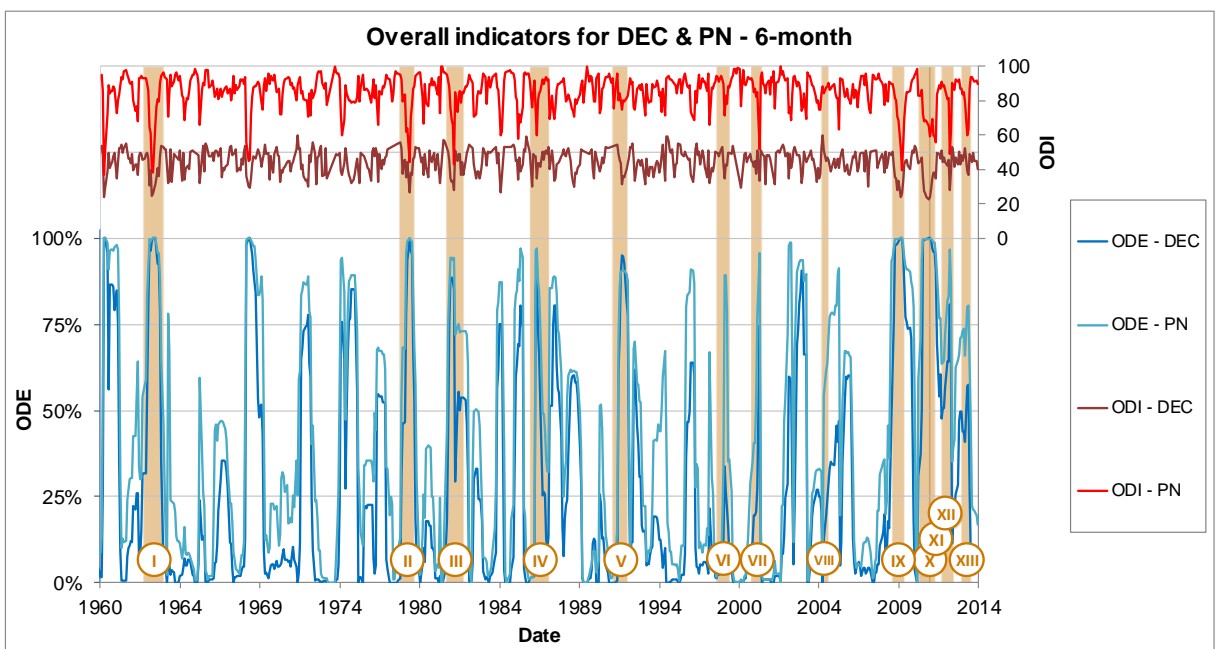

**Figure A6.** ODE and ODI values using the 6-month time scales of DEC and PN indices, compared with the 13 detected historical droughts (in orange).

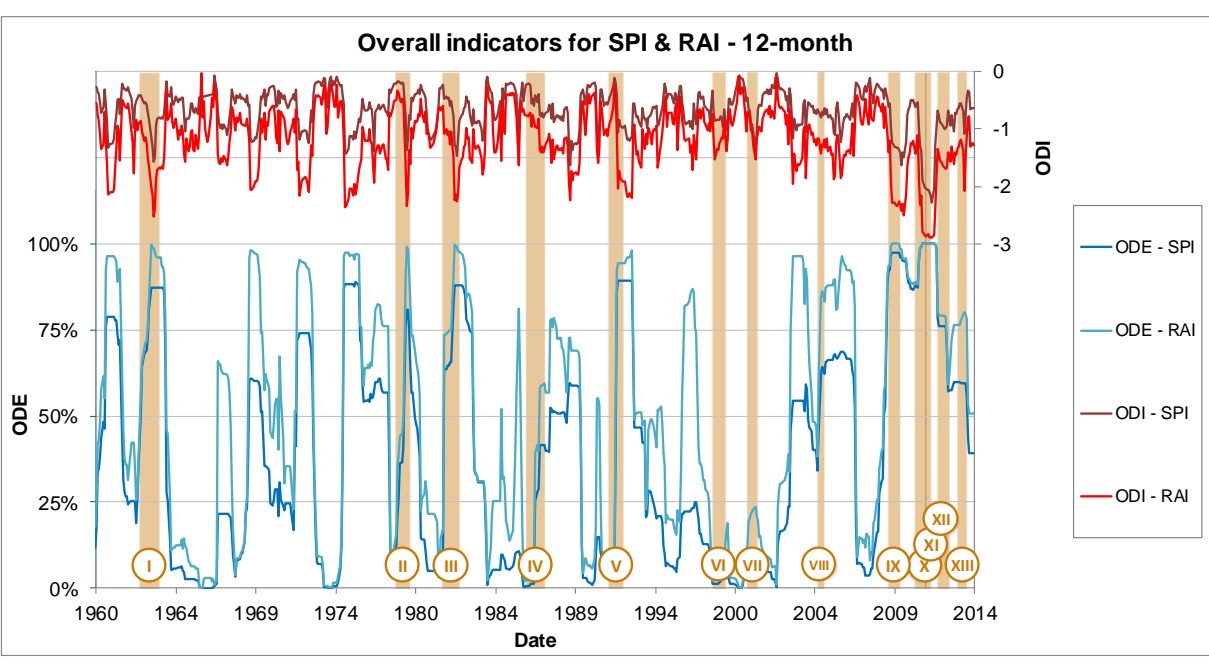

**Figure A7.** ODE and ODI values using the 12-month time scales of SPI and RAI indices, compared with the 13 detected historical droughts (in orange).

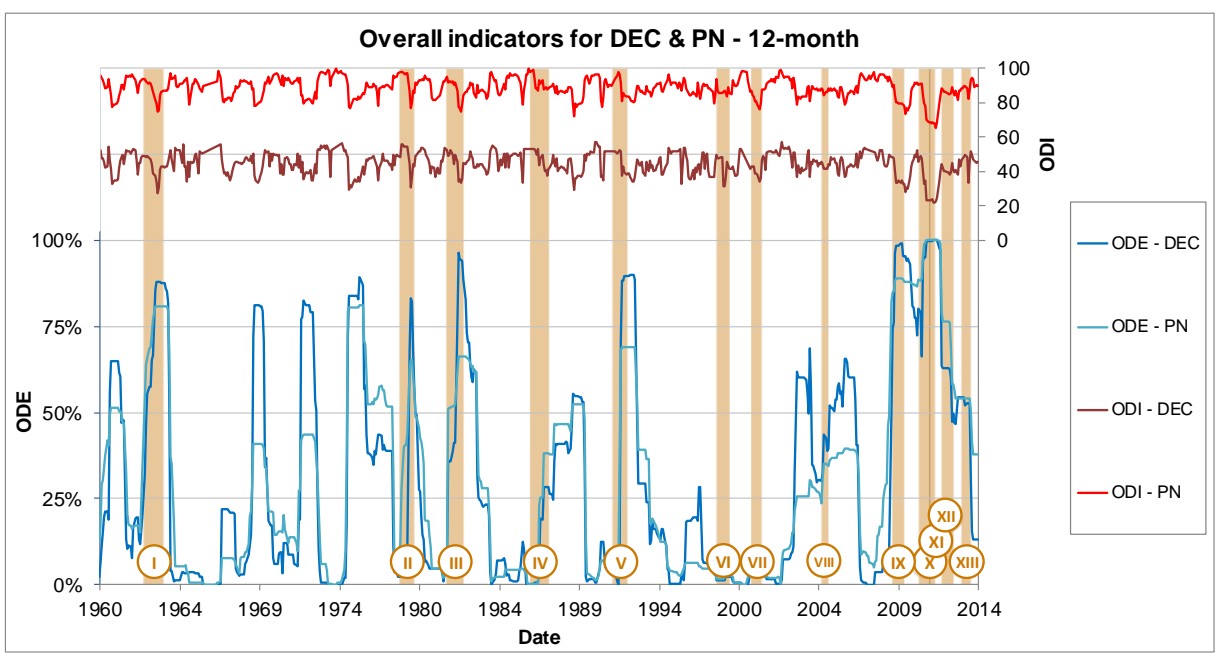

**Figure A8.** ODE and ODI values using the 12-month time scales of DEC and PN indices, compared with the 13 detected historical droughts (in orange).

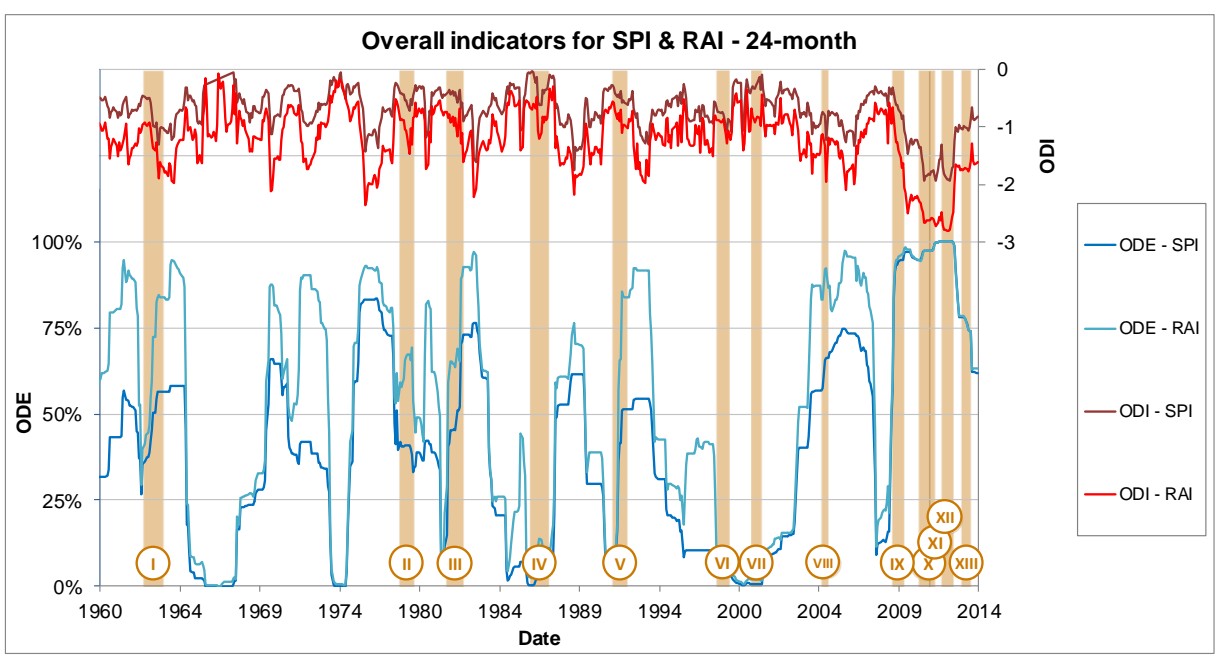

**Figure A9.** ODE and ODI values using the 24-month time scales of SPI and RAI indices, compared with the 13 detected historical droughts (in orange).

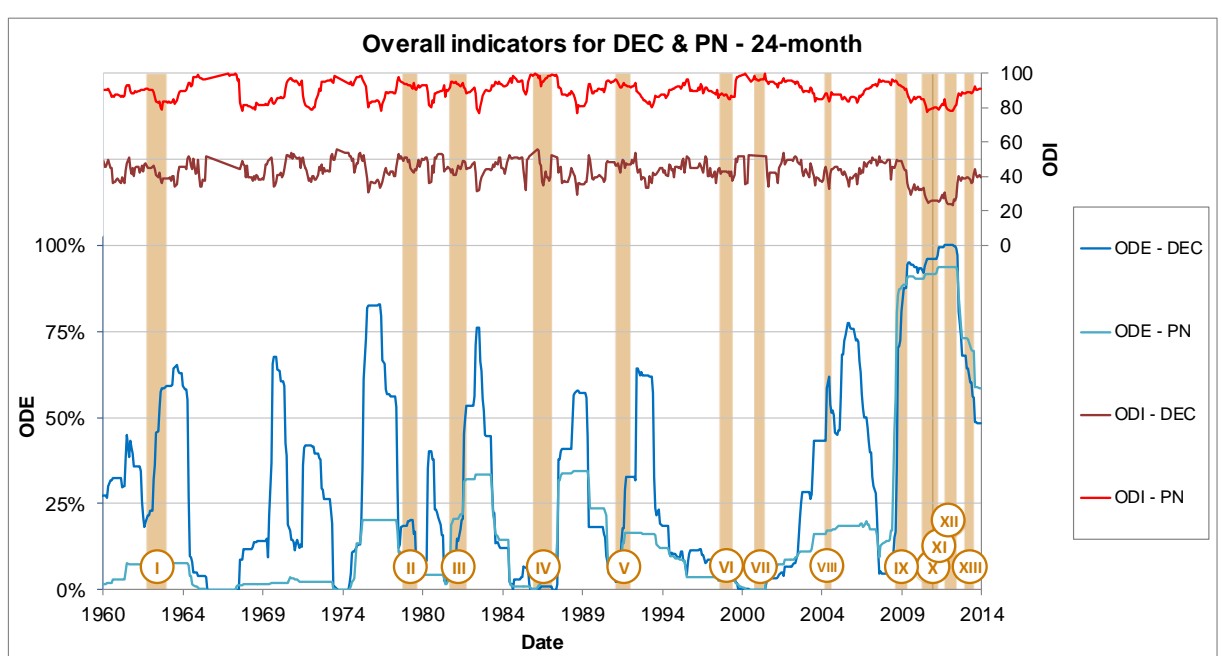

**Figure A10.** ODE and ODI values using the 24-month time scales of DEC and PN indices, compared with the 13 detected historical droughts (in orange).

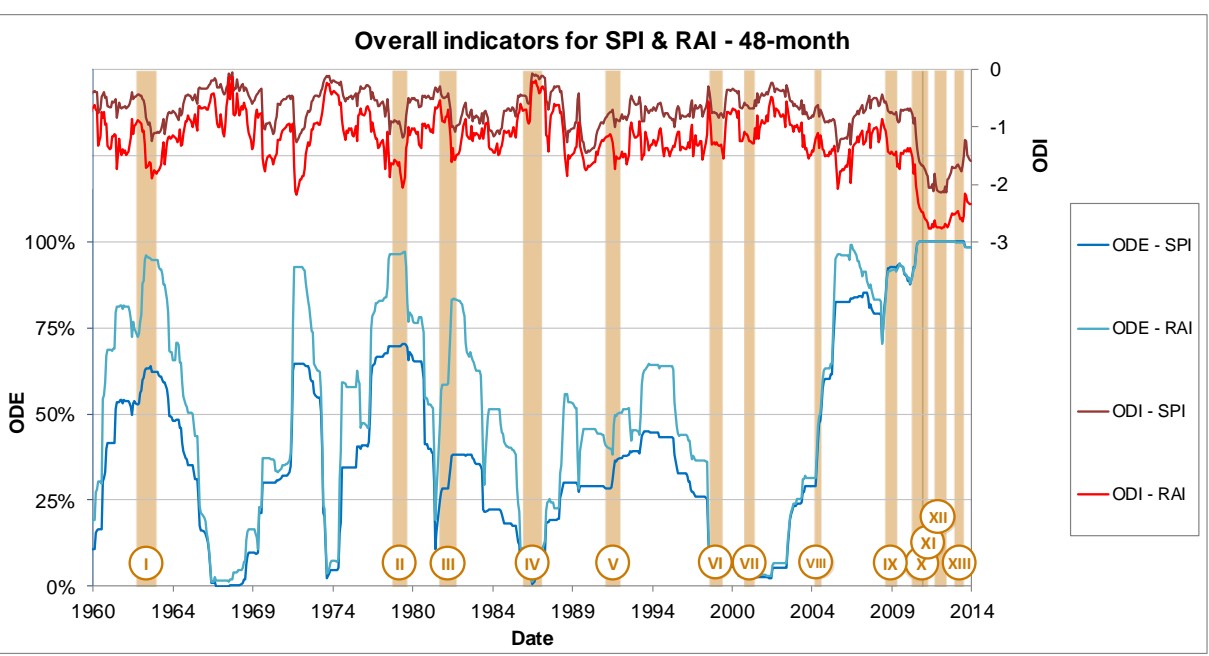

5  **Figure A11.** ODE and ODI values using the 48-month time scales of SPI and RAI indices, compared with the 13 detected historical droughts (in orange).

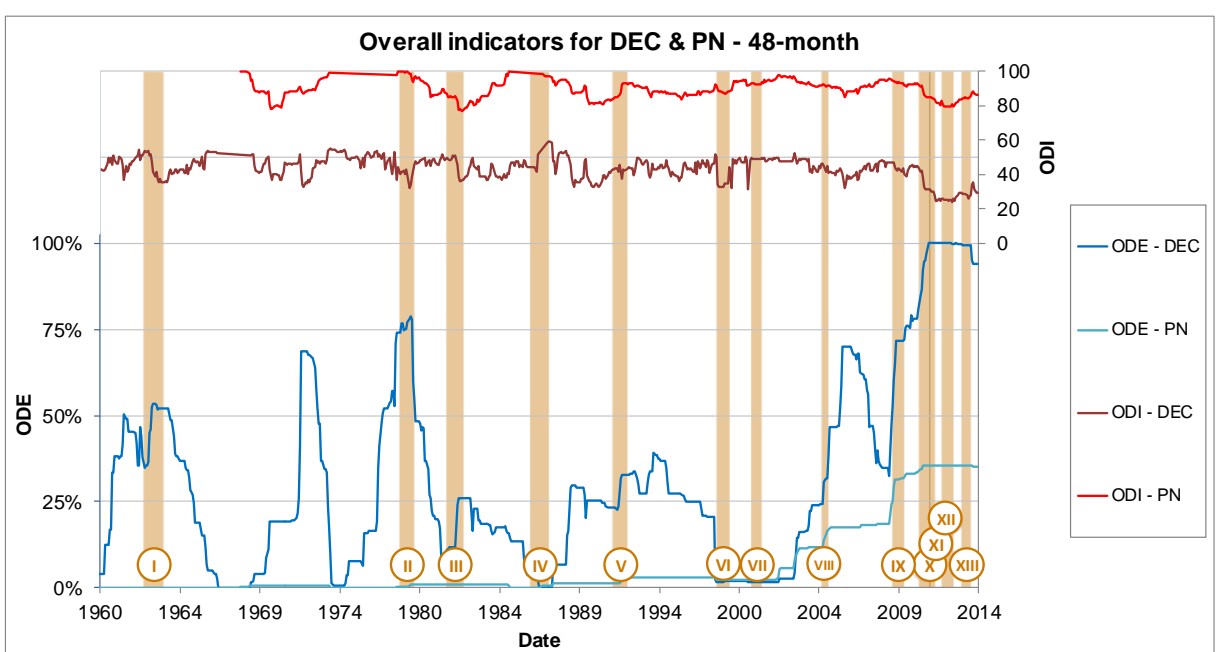

**Figure A12.** ODE and ODI values using the 48-month time scales of DEC and PN indices, compared with the 13 detected historical droughts (in orange).

## Appendix B: PSS results

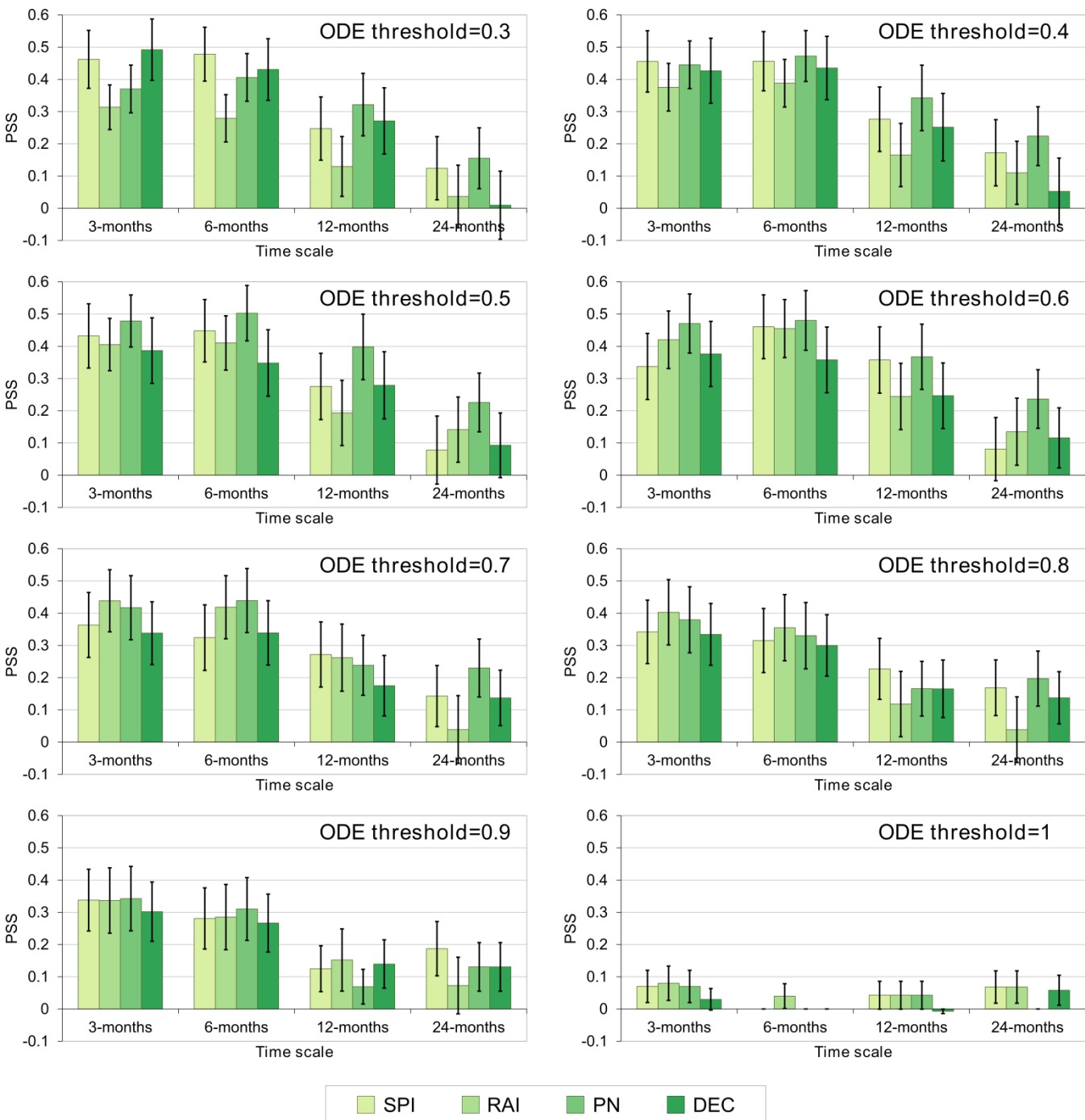

**Figure B1**: Graphical PSS results for different index and timescale combinations, for a range of ODE thresholds between 0.3 and 1. The black error bars represent the 95% confidence interval.

## Acknowledgments

This study is based on the project "Jinsha River Basin (JRB): lntegrated Water Resources and Risk Management under a Changing Climate" funded by the Swiss Agency for Development and Cooperation (SDC) and has been supported financially by the International science & Technology Cooperation Program of China (Grant No. 2014DFA71910).

The authors would like to acknowledge and thank the Bureau Of Hydrology (BOH) and the Changjiang River Scientific Research Institute (CRSRI) from the Changjiang Water Resources Commission for providing the data, and the enterprise Ernst Basler + Partner AG for the coordination and support throughout the entire project.

Comments by two anonymous reviewers and the journal editor are gratefully acknowledged.

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
