# Peer review of "Searching for the optimal drought index and time scale combination to detect drought: a case study from the lower Jinsha River Basin, China"

_Hydrology and Earth System Sciences, 2017_

## Author Response (AR1)

**AUTHOR'S RESPONSES**

These are the Authors' replies to comments from Referees #1 and #2. We use blue color for our replies and black color for Referee's comments.

It should be noted that the comments of the Referees make reference to pages and lines of the production paper. We use the same criteria in this reply.

**Author's response. Referee #1**

Summary: This manuscript presents an analysis of meteorological drought metrics over the lower Jinsha River Basin in China. They use precipitation data from 29 meteorological stations and calculate various formulations of the Standardized Precipitation Index (SPI), the Rainfall Anomaly Index (RAI), the Percent of Normal precipitation (PM) and Deciles (DEC). These indices are then evaluated spatially and in the context of their intensity using what the authors call the Overall Drought Extension (ODE) and the Overall Drought Intensity (ODI). Characterizations of these metrics are then compared to historical documentation of droughts over the lower Jinsha Basin from 1960-2014 to assess their efficacy in characterizing historical drought events. The authors make various conclusions about which of the indices and their spatial characterizations best represent the historical data.

General Remarks: This is generally a well written paper (it nevertheless could benefit from some English writing improvements and attention to typos throughout) that seeks to evaluate how best to characterize meteorological droughts over the lower Jinsha Basin. It should be published after some major revisions regarding clarity and content, which I outline as general comments below.

We want to thank Referee #1 for the General Remarks. English writing improvements will be carried out before submission of the revised manuscript. Aspects regarding clarity and content are treated below.

2. The authors make clear that their assessment is specific to meteorological drought and therefore focus exclusively on precipitation. This is fine as far as it goes, but they make several statements about temperature and river discharge assessments in the context of droughts that are too critical and not entirely accurate.

The authors wanted to stress that, in this particular case, precipitation data is the most reliable source of information. The questionable statements about temperature and discharge will be reviewed and adjusted.

Moreover, they point out that temperature/ET plays an important role in droughts within their study region (e.g. Pg. 3, Lns. 27-29). While they note as a caveat in their conclusions that ET has not been considered and may explain some of the deficiencies in their assessments, it is too little and too late in my opinion. The authors need to take on this obvious criticism of their study more directly and provide more guidance on how it might impact their results, if not try to quantify the impact of ET in an assessment metric. They also should not be so dismissive of the vast amount of work that has shown integrated drought metrics like modeled soil moisture, PDSI, SPEI, etc. to work as a suitable measure of drought (they mentioned the US Drought Monitor, but fail to note it is based on PDSI!). For instance, their paragraph starting on Pg. 2, Ln. 31 is far too dismissive of integrated metrics and

reads like a poor justification for why they focus only on precipitation. If they only have reliable precipitation data over their study region that is fine, but a focus on precipitation alone in this case should not be falsely justified by an attempt to dismiss integrated metrics. This aspect of the paper needs to be modified throughout.

We have only used precipitation-based indices for several reasons:

- The availability of measured meteorological data was limited; precipitation was found as the single most reliable type of information.
- It is true that the use of integrated drought metrics such as PDSI or SPEI could improve the scope and quality of the study and enrich the procedure. However, potential evapotranspiration (PET) data is required to compute these indices, and no reliable PET data was available for the study region. PET calculation depends on solar and longwave radiation, temperature, wind speed, and humidity. Although approximations may be used to estimate this variable, for example by only using temperature data, some studies (Jeevananda Reddy, 1995; Shaw and Riha, 2011; Staage et al., 2014) showed a high sensitivity of the PET to the chosen approximation method. A deeper analysis that helps selecting and applying such methods is needed.
- While this study is specific for the lower Jinsha River Basin, the procedure proposed is intended to serve as a basis for further studies in other regions where only precipitation data is available. This study should be seen as a test for other cases to validate whether precipitation-based indices can be used to predict droughts at a basin scale.

The authors will present these considerations more clearly in the manuscript to justify the only use of precipitation-based indices.

Same answer applies to the corresponding comment of Referee #2.

2. I am not convinced that the metrics proposed by the authors are new. They claim that the ODE and ODI are newly developed metrics and tout their development at multiple points within the manuscript. The ODE is just a form of drought area index and is no more than a measure of the total area of their study region in drought. A similar criticism can be made of the ODI. I therefore have no criticism of the application of these methods, just that they should not be touted as newly developed metrics or metrics of particular novelty that somehow add to the importance of their study.

As indicated by Referee #1 (and also Referee #2), some works (Bhalme and Mooley, 1980; Fleig et al., 2011; Mitchell et al., 1979) have already developed and used drought area indices, although without specifically using the SPI, RAI, PN and DEC indices for their definition. Consequently, the authors will mention these references in the manuscript and avoid presenting the ODE and ODI indices as newly developed. Instead, they will indicate that these indices (ODE and ODI) are an adaptation of existing ones.

3. The authors present quantitative metrics for comparing drought conditions based on their metrics and the historical records of droughts in the region. What is not clear, however, is how they actually translate the historical data into quantitative measures that can be compared to the drought metrics. In other words, they define skill scores in terms of hits, misses, etc., but what actually constitutes a hit or a miss? Is it just timing? Are magnitudes considered?

For the original paper, the authors had considered the temporal coincidence (timing) of drought events as defined by the ODE indices surpassing a predefined threshold (magnitude). The following definitions were used:

- A hit: when one (or more) drought detected according to the ODE values happened during the same year of a historical drought.
- A miss: when, during a year where a drought has been recorded, no event has been detected.
- A false alarm: when one (or more) drought detected according to the ODE values happened during a year when no event has been detected.

- A correct rejection: when, during a year where no droughts have been recorded, no drought has been detected according to the ODE values.

The maximum number of hits (or misses) was limited to the number of years of the study period, i.e. 55, which impacted on the confidence interval of the PSS and thus the precision of the PSS-based results.

The comments of Referee #1 have entailed a discussion among the authors of potentials for improving this approach. We now propose using a discretization by months (instead of by years) for the matching between detected and recorded droughts. That means that:

- We create 2 **monthly** series of events, each month being either "drought" or "non-drought": one series for the historical events; and one series for the detected events.
- For each month, we check if a drought in the detected series corresponds to a drought in the historical series, thus defining the hits, misses, etc.
- In this way, we increase the number of possible hits, misses… thereby increasing the sample size and reducing the confidence interval of the PSS.

Moreover, instead of calibrating different ODE thresholds for the different index-timescale combinations, we will now use the same threshold across all index/timescale combinations for the analysis, investigating the sensitivities of our results for a range of thresholds. This decision is based on the fact that ODE captures the coverage of droughts at the basin scale, which should not depend on the type of index used. We think this new approach is less arbitrary and more consistent than the original one. It is worth mentioning that new results have been obtained, which differ in some way from those presented in the original manuscript. In particular:

- The confidence intervals corresponding to each PSS value have been reduced, which implies a greater statistical confidence on the new results.
- In general the 3- and 6-month timescales offer better results than the 12- and 24-month timescales, practically for all the thresholds, while in the original work best results were found for the 6- and 12-month timescales. Figure 1 shows an example of results for 2 different ODE thresholds (0.45 and 0.8).
- Based on the PSS values and taking into account their confidence interval, there are no statistically significant differences of results across the different indices for the 3- and 6-month timescales. This indicates that indices perform similarly well, consistent with the fact that they all rely on the same type of data (precipitation).

A complete description of this new approach and the results obtained will be included in the final version of the manuscript.

[Figure]

[Figure]

*Figure 1. New PSS results. Comparison between ODE thresholds of 0.45 and 0.8.*

Note that drought magnitudes are not directly considered for the comparison between historical records and our metrics.

It is also not clear why the authors consider the historical accounts a reliable benchmark, relative to the more quantitative measure of droughts that they develop in their study. I do not think enough emphasis is placed on skill scores that are impacted by inaccuracies in the historical records (in terms of how well they characterize the timing, severity and spatial extent of droughts) relative to what the authors construct from the network of precipitation records.

The catalogue of historical droughts is mainly used to find which combination of index and timescale best fit these catalogued events for different thresholds of the ODE indicator. That means we considered that the information collected during the compilation process is the basis to which quantitatively detected droughts must be compared, simply because no other benchmark exists in this region. While the historical accounts may not be entirely reliable, in this study they are used as a reference for lack of better information.

However, as mentioned in the manuscript, historical accounts should be addressed carefully, in particular regarding the reliability of the data sources and their ambiguity. The compilation process of information relative to historic drought events will be described in more detail in the manuscript. Particular focus will be set on the type and form of information which was available and used. And the authors will discuss the availability and accuracy of the information on the drought characteristics (such as date, duration, area, etc.).

The authors will discuss the expected sensitivity of the results to the historical records.

Specific Comments:

Pg. 10, Ln. 17: While not essential, the authors might consider using two consecutive positive or negative years to end or start a drought. There are definitely periods in Figure 2 that identify droughts as separated by a single year of positive SPI values or very short droughts that represent just single-year excursions. If more persistent and widespread droughts are the interest, a 2-yr criterion for beginning and ending droughts might help.

Indeed, Figure 2 shows very short drought events (1-2 months). This is more noticeable for low timescales and is due to the identification criteria based on the index values (e.g., the criteria defined by McKee et al. (1993) for the SPI). The authors agree with Referee #1 and consider that it would be convenient to set a minimum duration of droughts. According to the range of drought durations of the historical events recorded (≥3months and ≤13months), a **2-year** criterion for beginning and ending droughts seems too restrictive. The authors will adapt the applied identification criteria to avoid an overestimation of events by defining a minimum duration of dry (and wet) periods of 3 months.

Pg. 15, Ln. 6: The authors optimize the characteristics of their drought metrics based on skill assessments over the full historical interval. This is akin to calibrating the forecast model and then performing in-sample skill assessments. A more rigorous assessment would be to optimize over a specific period and then assess the skill in an out-of-sample period. This could be done using block hold out periods or leave half out assessments. As it stands, however, the authors optimize over the same period that they assess the skill of their metrics. This is particularly relevant when considering the authors' methods for future drought assessments. Their in-sample skill assessment is very likely to exaggerate the efficacy of their metrics for future droughts.

The main objective of our work is to identify which combination of index and timescale offers good correlation with the observed events. The fact that only 13 events have been documented is an important limitation, and the authors realize that an optimization of the ODE threshold based on this limited sample is not robust. Any meaningful cross-validation of this optimization would require a larger sample. Instead, the authors no longer search for an optimal threshold but explore the effect of varying the threshold in a reasonable range. Since the general findings turn out to be independent of the specific threshold, we consider them robust.

Same answer applies to the corresponding comment of Referee #2.

**Author's response. Referee #2**

General comments

This is an interesting paper thanks to the region of interest, the compilation of historical documents, the use of station data and the comparison of various drought indices.

We want to thank Referee #2 for the General comments.

Specific comments

While the ODE and ODI indexes proposed are indeed relevant in this context, I don't think they can be presented as extremely innovative, as similar indices have been used to study drought area and intensity.

As indicated by Referee #2 (and also Referee #1), some works (Bhalme and Mooley, 1980; Fleig et al., 2011; Mitchell et al., 1979) have already developed and used drought area indices, although without specifically using the SPI, RAI, PN and DEC indices for their definition. Consequently, the authors will mention these references in the manuscript and avoid presenting the ODE and ODI indices as newly developed. Instead, they will indicate that these indices (ODE and ODI) are an adaptation of existing ones.

Second, a big caveat is that the skill scores are computed on the same period than the one chosen to determine the ODE thresholds for detection. The fact that only 13 events have been documented is an understandable limitation; however, this method will likely create an overestimation of the power of the index to detect droughts. A more rigorous "cross-validation" procedure is needed (e.g. segmenting the record period and perform the study leaving one segment out each time?).

The fact that only 13 events have been documented is indeed an important limitation and the authors realize that an optimization of the ODE threshold based on this limited sample is not robust. Any meaningful cross-validation of this optimization would require a larger sample. Instead, the authors no longer search for an optimal threshold but explore the effect of varying the threshold in a reasonable range. Since the general findings turn out to be independent of the specific threshold, we consider them robust.

Same answer applies to the corresponding comment of Referee #1.

Moreover, while it is absolutely true that drought measures such as the SPEI and PDSI have shortcomings – in particular the reliance on PET rather than ET, they do capture features that precipitation-only indices cannot see. It is absolutely fine if data is not available to compute such indices, but it should be the main reason for not comparing what these other indices would say relative to the historical data. It may not be very useful, but I wonder if global PDSI/SPEI datasets capture anything in that region during the drought events mentioned (even if they have a much lower resolution).

We have only used precipitation-based indices for several reasons:

- The availability of measured meteorological data was limited; precipitation was found as the single most reliable type of information.
- It is true that the use of integrated drought metrics such as PDSI or SPEI could improve the scope and quality of the study and enrich the procedure. However, potential evapotranspiration (PET) data is required to compute these indices, and no reliable PET data was available for the study region. PET calculation depends on solar and longwave radiation, temperature, wind speed, and humidity. Although approximations may be used to estimate this variable, for example by only

using temperature data, some studies(Jeevananda Reddy, 1995; Shaw and Riha, 2011; Staage et al., 2014) showed a high sensitivity of the PET to the chosen equation. A deeper analysis that helps selecting and applying such methods should be performed.

- While this study is specific for the lower Jinsha River Basin, the procedure proposed is intended to serve as a basis for further studies in other regions where only precipitation data is available. This study should be seen as a test for other cases to validate whether precipitation-based indices can be used to predict droughts at a basin scale.

As already suspected by Referee #1, we have not used PDSI/SPEI datasets because of their too low resolution for properly capturing the spatial detail of the events.

The authors will present these considerations more clearly in the manuscript to justify the only use of precipitation-based indices.

Same answer applies to the corresponding comment of Referee #1.

Furthermore, for clarity, it may be useful for the authors to develop a little more the compilation process of documents relative to drought in the paper itself, and explain in a little more detail why they consider that the spatial distribution of the stations and the quality of the records are good enough for the study they want to perform. It would also be nice if the question the bias introduced by station locations was treated with more detail. Related to this, how was the grid resolution chosen (p.12)?

The compilation process of information relative to historic drought events will be described in more detail in the manuscript. Particular focus will be set on the type and form of information which was available and used. And the authors will discuss the availability and accuracy of the information on the drought characteristics (such as date, duration, area, etc.).

We consider that the spatial distribution of the stations is adequate for the purposes of the study: stations are distributed more or less evenly both in the X- and in the Y-axis. There are no zones with a significantly denser presence of stations that could overestimate their importance.

Regarding the quality of the records, it should be mentioned that all the precipitation data used have been provided by the China Meteorological Administration (CMA) and downloaded from the "China Meteorological Data Sharing Service System" (http://cdc.nmic.cn/gx/web/yqlj.jsp). A preliminary quality check and correction of datasets (including data gap-filling) was already done by CMA before uploading them to the system.

Regarding the grid resolution, we have chosen a 400x300 cells grid as a trade-off between the density of points and the computational requirements. The grid density used corresponds more or less to 1 cell/3.2 km$^2$, which is certainly adequate for the purposes of the study. However, this choice must be adapted to the needs of potential other cases: computation time, data availability, variations of precipitation patterns, changing topography, etc.

Finally, the sensitivity of the ODE thresholds chosen to the classifications proposed in table 6 and to the definition of the beginning and end of droughts should be discussed briefly.

Authors will include a summary of the sensitivity analysis performed on the ODE thresholds that will show that their variation hardly affects the overall findings on the best performing index/timescale combinations.

Concerning the thresholds of Table 6, these are based on standard criteria (Jain et al., 2015; McKee et al., 1993; Tsakiris et al., 2007) and supported by follow-up literature. A sensitivity analysis of these particular thresholds could be interesting for a supplementary work but may exceed the scope of this paper.

Technical corrections

I have found that the paper should undergo significant editing. However, I am not a native English speaker myself you may not want to follow exactly the suggestions given below. In the following, I suggest replacements:

English writing improvements will be carried out before submission of the revised manuscript. All further technical corrections of Referee #2 will be taken into account in this process.

p.1:

l.25: "Historical drought events which occurred"?

l.27" "that best reproduce"

p.2:

l.7 : "in agriculture" by "to the agriculture sector"?

l.16: "that is the case of" by "an example is"?

l.17: "the China's National Climate Change" by "China's National Development and Reform Commission"?

l. 26: "Main advantages" by "the main advantages"?, "the ease of use" by "their ease of use", "the limited need of data " by "the limited data requirements",

l.27: "capacity to an early detection of drought events" by "capacity for early detection of drought events"?

l.33: "is depending on" by "depends on"

p3:

l.2: "more exhaustive work" "more time-consuming work"

l.4 "This allows identifying" by "This enables on to the identify"

l.10: "do not imply necessarily" by "do not necessarily imply"

l.23-24 "fall" by "discharge"?

l.31 "are susceptible to be affected" by "can be affected"?

p.4:

l.3 'location" by "locations"

p.5

l.8: "For the last 20 years, detailed information is available regarding all drought events" by "Detailed information is available for all drought events over the past 20 years"

l. 18: "The use of meteorological indices allows analyzing the influence" "allows one to analyze"

p.7

l.5: "This allows characterizing" "This allows for the characterization of... and thus facilitates"

l. 6: "each station surroundings" by "each station's surroundings"

l.19: "This allows defining" by "This allows us to define"

l.25: "it helps defining" by "it helps define"

p.13

l.8: "completing the collected historical records for little information regarding the magnitude of the events has been found" by "complete the collected historical records which include little information on the magnitude of the events

l. 9 "Not defined values" by "Undefined values"

l.11: "On purpose, only cells under drought conditions have been considered for the definition of this indicator by" Only cells under drought conditions have been considered to define this indicator" "If the ODI was calculated as an average value for the entire basin (as adopted for instance in Trambauer et al. 2014)) higher (or lower) indicator values in a part of the basin may compensate lower (or higher, respectively) indicator values in the rest of the basin, offering an overall value close to normal precipitation." by "If the ODI had been calculated as an average value for the entire basin (as adopted for instance in Trambauer et al. 2014)) higher (or lower) indicator values in a part of the basin may have compensated for lower (or higher, respectively) indicator values in the rest of the basin, yielding an overall value close to normal precipitation."

p.14

l23-24: "have been" by "were"

p.15

l.3-4: idem

p.17

l.1: "that correspond with" by "that correspond to"

l.7: "the droughts occurred" by "the droughts which occurred"

p.19:

l.1: "higher" by "highest"

l.2: "false positives" by "false positive"

l.4: "quite" by "well"

l.5: "in relation" by "in comparison"

l.7: 'droughts have been chronicled" by "drought has been chronicled"

l.10: 'not wide' by 'spatially concentrated'

l.12: 'have been' by 'were'

l.13 'identifying these events is possible, although it is difficult to disentangle them"

l.16: 'by the use' by 'using'

l.27: "some considerations are recommended" by "caution is advised"

l.29: delete "some", "proved by 'proven'

l.30: "The variability of temperature, for instance, may have an important impact on the crop water availability and then in the assessment of agricultural droughts, although it has not been taken into account" by "temperature variability, not considered here, can play a significant role in the onset of agricultural drought"

p.20

l.10: "This work represents an attempt at building a tool..."

l.13: "was compiled"

l.14: "were identified and catalogued"

l.23"indexes and time scales"

l.25" "consecutive or "clustered in time' rather than "more consecutive"

l.28" supposes" by "represents"

l.32" facing" by "for"

**Searching for the optimal drought index and time scale combination to detect drought: a case study from the lower Jinsha River Basin, China**

Javier Fluixá-Sanmartín[1], Deng Pan[2], Luzia Fischer[3], Boris Orlowsky[4], Javier García-Hernández[1], Frédéric Jordan[5], Christoph Haemmig[3], Fangwei Zhang[6], Jijun Xu[2]

[1]Centre de Recherche sur l'Environnement Alpin (CREALP), Sion, 1951, Switzerland
[2]Changjiang River Scientific Research Institute, Changjiang Water Resources Commission, Wuhan Hubei, 430010, China
[3]GEOTEST AG, Zollikofen, 3052, Switzerland
[4]cClimate-babel, Zurich, 8047, Switzerland
[5]Hydrique Ingénieurs, Le Mont-sur-Lausanne, 1052, Switzerland
[6]Bureau of Hydrology, Changjiang Water Resources Commission, Wuhan Hubei, 430017, China

*Correspondence to*: Javier Fluixá-Sanmartín (javier.fluixa@crealp.vs.ch)

**Abstract.** Drought indices based on precipitation are commonly used to identify and characterize droughts. Due to the general complexity of droughts, the comparison of index-identified events with droughts at different levels of the complete system, including(e.g., soil humidity or river discharges), rely typically on model simulations of the latter, entailing potentially significant uncertainties.

The present study explores the potential of using precipitation based indices to reproduce observed droughts in the lower part of the Jinsha River Basin, proposing an innovative approach for a catchment-wide drought detection and characterization. Two new indicators, namely the Overall Drought Extension (ODE) and the Overall Drought Indicator (ODI), have been developeddefined. These indicators aim at identifying and characterizing drought events at basin scale, using results from four meteorological drought indices (Standardized Precipitation Index, SPI; Rainfall Anomaly Index, RAI; Percent of Normal precipitation, PN; Deciles, DEC) calculated at different locations of the basin and for different time scales. Collected historical information on drought events is used to contrast results obtained with the indicators.

This method has been successfully applied to the lower Jinsha River Basin, in China, a region prone to frequent and severe droughts. Historical drought events occurred from 1960 to 2014 have been compiled and catalogued from different sources, in a challenging process. The analysis of the newly developed indicators shows a good agreement with the recorded historical drought events at basin scale. It has been found that the combinations of index and time scale that best reproduces observed events across all the indices is the 6-month time scaleare the SPI-12 and PN-12 for long droughts (1 year or more) and the RAI-6, PN-6 and DEC-6 for shorter or more consecutive events.

[revised manuscript text omitted]
 have beenwere identified based on the indexse values. Then, the ODE and ODI indicators have beenwere calculated for the lower JRB. An example of tThe resulting ODE and ODI series are is shown in Figure 4 and Figure 5 for the SPI-6 and RAI-6 combinations andas well as in Appendix A for all the time scales and indices analyzed, along with the recorded historical droughts shaded in orange.

The objective is to establish a combination of time scale and index that offers an optimum identification of historical droughts. As stated before, the main criteria used to contrast the performance of the forecasts is that a drought event is supposed to happen when the ODE value exceeds a threshold that is to be defined for each combination. The combination finally retained should maximize the number of hits and minimize the misses between the forecasts and the observed events.

[Figure]

**Figure 3.** Extrapolated SPI-12 values in  October 2013 for the entire lower JRB.

[Figure]

**Figure 4.** ODE and ODI values using the 6-month time scales of SPI and RAI indices, compared with the 13 detected historical droughts (in orange).

[Figure]

**Figure 5.**

The 1-month scale overall indices show rapid fluctuations that correspond to short periods of precipitation deficiency not captured in the catalogue of historical droughts. This is mainly due to punctual large rainfall events, that have an important influence in the indices which may indicate that the drought had ceased when it is not the case (Barua et al., 2011). The use of this time scale is not recommended for drought monitoring since long drought events are hardly identified. The opposite effect occurs when using the 48-month scale. The inertia of the rainfall shortage tendencies may mask shorter droughts and overestimate their durations. Since most of the episodes last one year or less (Table 1), they are hardly detected using the 48-month scale. The droughts which occurred from 2009 to 2014 (droughts IX to XIII) illustrate this phenomenon: even if five different droughts have been catalogued, a unique one is detected using the 48-month scale, according to the ODE time series. Therefore, using the 1- and 48-month scales do not provide any substantial information about the occurrence and duration of the droughts and have been excluded from the performance analysis.

For the rest of the time scales (3-, 6-, 12- and 24-month), the ODE threshold indicating the occurrence of a drought  is required for the computing of the PSS that will serve as a support for the selection of the best combination of index and time scale. Traditionally, cross-validation techniques are used to define optimum thresholds, for when within a training subset the threshold maximizing the PSS is identified and validated in a non-overlapping validation subset. The limited number of 13 independent events in our record prevents following this approach. Instead, a sensitivity analysis was performed using the same threshold across all of the combinations and exploring the effect of varying it in a reasonable range (in this case, from

0.3 to 1 by 0.1 steps). The resulting PSS values are shown in Figure B1 of Annex B along with the 95% confidence interval, which allows indicating whether the score is significantly different from zero.

Based on the resulting ODE series, a set of thresholds has been manually estimated () to match as many observed events as possible. According to these thresholds, the PSS is calculated for each index and time scale combination () along with its 95% confidence interval, which allows indicating whether the score is significantly different from zero as stated above. Graphic results are presented in Figure 5.

**Table 8.** Thresholds estimated for the ODE indicating the occurrence of a drought.

| Index | Time scale | | | |
|---|---|---|---|---|
| | 3-month | 6-month | 12-month | 24-month |
| SPI | 75% | 75% | 65% | 50% |
| RAI | 95% | 85% | 75% | 75% |
| PN | 95% | 85% | 60% | 25% |
| DEC | 75% | 75% | 75% | 60% |

**Table 9.** Peirce skill score values for each combination of index and time scale, with 95% confidence intervals.

| Index | Time scale | | | |
|---|---|---|---|---|
| | 3-month | 6-month | 12-month | 24-month |
| SPI | 0.34 ± 0.18 | 0.23 ± 0.18 | 0.38 ± 0.18 | 0.17 ± 0.18 |
| RAI | 0.3 ± 0.16 | 0.4 ± 0.14 | 0.25 ± 0.18 | 0.17 ± 0.18 |
| PN | 0.37 ± 0.17 | 0.44 ± 0.16 | 0.45 ± 0.16 | 0.32 ± 0.16 |
| DEC | 0.29 ± 0.18 | 0.38 ± 0.18 | 0.32 ± 0.17 | 0.25 ± 0.18 |

[Figure]

**Figure 5.** Graphic representation of the PSS results for an ODE threshold of 0.4, with the black error bars representing the 95% confidence interval (±1.96 standard errors).

Most of the 95% confidence intervals of the PSS do not include zero, disproving that skill scores could have identified drought events by chance sampling fluctuations. On the contraryOnly, for the SPI and RAIsome of the indices at 24-month (e.g., RAI-24 for an ODE threshold=0.7), results cannot assert that skill scores are significantly different from zero and thus these two combinations should not be considered.

Attending to the PSS values (Figure B1), results show a consistent tendency across all ODE thresholds of higher PSS at the 3- and the 6-month time scales. An example of PSS for an ODE threshold of 0.4 is presented in Figure 5. Moreover, there is no single index that clearly produces better results. Indeed, based on the PSS values and taking into account their confidence intervals, there are no statistically significant differences across the different indices for the 3- and 6-month time scales. This indicates that, for these time scales, all the indices perform similarly well on capturing the events, which is consistent with the fact that they all rely on the same type of data (precipitation). PSS results are independent of the specific threshold and thus they are considered robust. However, it is worth mentioning that in general, higher PSS for the 3- and the 6-month time scales are produced using ODE thresholds between 0.4 and 0.6.

In general, the 6- and 12-month time scales shows a better performance on detecting historical droughts, in particular when using the PN index. It is thus recommended to use the 12-month indices to assess drought occurrence of one year duration, and the 6-month indices for shorter or more consecutive events (e.g., droughts IX to XIII).

Attending toRegarding the 126-month ODE series (Figure A5 and Figure A6 of Appendix AFigure 4), it is important to highlight some relevant aspects:

- All the observed drought events have their corresponding ODE series have peaks corresponding to the drought events I, II, III and V.
- The drought event IV is captured by an increase of the ODE values. This increase is shifted forward, starting in the middle of the drought event IV and having its peak around 1990 (around 3 years later than specified in the catalogue). Nevertheless, as indicated in the catalogue of droughts (Table 1) the exact start date and duration of this event are unknown and could have occurred later.
- Although event VIII has an estimated duration of 3 months, ODE and ODI results consistently show a longer drought occurring in two phases (two consecutive increases of their values), covering a period of 1.5 and 2.5 years respectively. Event VIII seems to correspond to the second of these phases. Again, Tthe exact period of this drought is not well defined as indicated in the catalogue, leaving room for a longer duration of the real episode.
- Among the four different meteorological indices, the RAI presents the higher variability which may lead to inconsistencies with the catalogued droughts. A clear example is the false positives detected in 1997 that does not correspond with any recorded event.
- In general, the SPI and the DECall the indices are quitewell correlated, identifying most of the recorded droughts. The PN index behaves similarly, although it tends to underestimate the ODE values in relation to the SPI and the DEC.
- Three Several droughts of increasing magnitude are consistently detected between event I (1962) and II (1979) even if no droughts havedrought has been chronicled (false alarms). This may correspond to the above-mentioned scarcity of reliable information on droughts prior to 1980.

- While two episodes are reported in 1999 (event VI) and in 2001 (event VII), no ODE has captured them. However, the SPI based ODI shows a significant decrease corresponding to event VII, which may indicate a not wide but intense drought.
- As mentioned above, during the period 2009-2014, five consecutive events (IX, X, XI, XII and XIII) have been reported. Using the 12 month series a certain spotting of these events can be achieved, although it tends to aggregate them in one or two unique episodes.

For the 6 month time scale (Figure A5 and Figure A6 of Appendix A):

- The drought events IX, X, XI, XII and XIII droughts are well captured by the use of the 6 month timescale. As shown in Figure A5 and Figure A6, the different events during this period (2009–2014) match with the consecutive increases in the ODE values for all the indices (DEC, PN, RAI, SPI).

- However, the 6-month series of ODE suggest some false positive detections: more drought events than the observed are calculated. An overestimation of the influence of short periods of rainfall scarcity may be masking the true duration of the droughts.

Regarding the 3-month ODE series (Figure A3 and Figure A4), results suggest an overestimation of the number of detected events, as sometimes several detected events combine into one (longer) observed event. The 6-month time scale appears as more appropriate.

In general, the 1 , 3 , 24  and 48 month time scales do not reproduce the observed events and are not recommended. In summary, aAccording to the ODE series represented in Figure 4 and Figure 5 and in Appendix A, and to the forecast verification carried out with the Peirce skill score (Appendix BFigure 6), it seems that the best combination time scale for the identification of long droughts is the SPI or the PN indices at a 12at 6-months time scale. Results show an equally effective performance of the ODE series for all the indices, and the RAI, the PN or the DEC at a 6 month scale for shorter or more consecutive events. However, the risk of false positives must be addressed carefully, as the observation record likely misses events, in particular between 1962 and 1979, especially for 6 month scales.

Despite the good performance shown by the proposed overall indicator ODE to detect droughts, some considerations are recommendedcaution is advised. In particular, the choice of meteorological indices as a basis for the calculation of the ODE and ODI can lead to some errors when assessing drought occurrence. It has been proved that not all indices are equally capable of identifying droughts in this particular region. Temperature variability, not considered here, can play a significant role in the onset of agricultural droughtThe variability of temperature, for instance, may have an important impact on the crop water availability and then in the assessment of agricultural droughts, although it has not been taken into account. Besides, changes in the regulation infrastructures such as reservoirs have a growing influence on water supply. HenceM, meteorological indices aremay 
[revised manuscript text omitted]